# GENERATING SCENES WITH LATENT OBJECT MODELS

## ABSTRACT

We introduce a structured latent variable model that learns the underlying data-generating process for a dataset of scenes. Our goals are to obtain a compositional scene representation and to perform scene generation by modeling statistical relationships between scenes as well as between objects within a scene. We take inspiration from visual topic models and introduce an interpretable hierarchy of scene-level and object-level latent variables (i.e., slots). To generate scenes, dependencies between objects must be modeled. While we could achieve this by assuming slots are generated autoregressively, algorithms for estimating autoregressive slot posteriors impose an unnatural order on objects in a scene and are known to be limited in visually complex environments. We eliminate the need for autoregressive posterior inference by assuming that the assignment of objects to slots during generation is a deterministic function of the scene latent variable. We propose an inference algorithm that indirectly estimates an ordered slot posterior by leveraging an orderless slot posterior and the deterministically-predicted object generation order. Qualitative and quantitative analysis establishes that our approach successfully learns a smoothly traversable scene-level latent space and that the hierarchy of scene and slot variables improves the ability of slot-based models to generate scenes with complex multi-object relations.

## 1 INTRODUCTION

This paper considers the problem of learning the data-generating process for multi-object scenes. We want to train likelihood-based deep generative models that can encode scenes into a representation hierarchy of scene and object latent variables and that support novel scene generation. A scene-level representation can capture statistical relationships between scenes and can model complex interactions between objects (Jiang & Ahn, 2020). Object-level representations increase compositionality, interpretability, and robustness of the model to out-of-distribution data (Dittadi et al., 2021). Much progress has been made on models that encode scenes into sets of object-centric distributed representations, or *slots* (Greff et al., 2017; Burgess et al., 2019; Greff et al., 2019; Engelcke et al., 2019; Locatello et al., 2020; Nanbo et al., 2020; Emami et al., 2021; Engelcke et al., 2021). These models have demonstrated promising generalization and object-level disentanglement and have been applied successfully to object-centric visual model-based reinforcement learning (Veerapaneni et al., 2019).

However, training slot-based models to jointly decompose and generate novel scenes remains challenging. The limited prior work on this problem has approached it by either ignoring dependencies between slots (Greff et al., 2019; Emami et al., 2021) or by factorizing the joint distribution over slots using the chain rule into a sequence of autoregressive distributions (Engelcke et al., 2019; 2021). None of these models possess a scene-level representation and are therefore limited in their ability to capture scene-level statistics and higher-order object relations (Jiang & Ahn, 2020). Also, although autoregressive generation is capable of modeling dependencies between objects, *autoregressive inference* is extremely difficult (Engelcke et al., 2021) and remains an open problem.

In this work, we introduce Latent Object Models (LOMs), a deep generative modeling framework for fitting the data-generating process for scenes with a latent variable hierarchy consisting of a scene-level variable and a set of slot variables. The analogy used to design the hierarchical graphical model is inspired by visual topic models (Hofmann, 1999; Blei et al., 2003; Sivic et al., 2005; Fritz &

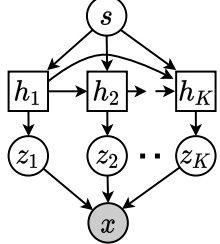

Figure 1: **The hierarchical graphical model for LOMs**. We assume that scenes $x$ are generated from $K$ object-centric slot latent variables $\mathbf{z}_{1:K}$. The slots are independent *conditional* on a causal sequence of deterministic variables (shown in boxes). This causal sequence is predicted from the top-level scene latent variable $\mathbf{s}$. By conditioning each slot on one of the deterministic variables, we provide the order-dependent relational information necessary for scene generation.

Table 1: Latent Object Models introduce the idea of using a scene-slot latent hierarchy to fit the data-generating process. Unlike spatial latents, slots are assigned to scene objects during inference and generation, improving interpretability and disentanglement of the learned representations.

| Model | scene latent | spatial latents | object slots |
|---|:---:|:---:|:---:|
| VAE (Kingma & Welling, 2013) | ✓ | | |
| NVAE (Vahdat & Kautz, 2020) | | ✓ | |
| GNM (Jiang & Ahn, 2020) | ✓ | ✓ | |
| EMORL (Emami et al., 2021) | | | ✓ |
| GENESIS (Engelcke et al., 2019) | | | ✓ |
| GENESIS-v2 (Engelcke et al., 2021) | | | ✓ |
| LOMs (Ours) | ✓ | | ✓ |

Schiele, 2008; Tuytelaars et al., 2010):

$$\text{documents} - \text{scenes}$$
$$\text{topics} - \text{slots}$$
$$\text{words} - \text{percepts}$$

where a percept is any discrete token that makes up a scene observation, such as a pixel or a 3D point. LOMs have a three-level hierarchy. Each percept is modeled as a $K$-component mixture, where each component is dependent on one of $K$ intermediate-level slots. Then, the slot variables are dependent on a single top-level scene variable. Although topic models make a "bag-of-words" assumption that topics are orderless within a document, multi-object scene generation requires modeling dependencies between slots. LOMs therefore cannot assume slots are orderless in a scene.

Instead of modeling slot generation with an autoregressive (i.e., stochastic) prior, which requires inferring an autoregressive posterior, we propose to assign objects to slots for ordered scene generation deterministically. We train a function to take a sample from the scene prior and predict a causal sequence of *deterministic* variables (Figure 1). The slot priors are designed to be independent of each other when conditioned on this sequence of deterministic variables, which eliminates the need to estimate an autoregressive posterior. We instead propose an inference algorithm for LOMs that assumes an auxiliary generative model which decomposes scenes into an orderless set of slots. This is used to infer a slot-order-invariant scene-level posterior from which we obtain an ordered slot prior. To obtain an ordered slot posterior for maximum likelihood training, we compute a matching between the means of the orderless posterior and the ordered prior. This matching is used to permute the orderless posterior and convert it into an ordered distribution.

**Contributions:** We introduce a latent variable model with a latent scene-slot hierarchy and variational inference algorithm for modeling the data-generating process of multi-object scenes. We demonstrate that this framework is capable of learning a low-dimensional scene-level manifold that can be smoothly traversed. We also show that generated scenes accurately capture higher-order object relations present in the dataset.

## 2 RELATED WORK

We compare LOMs with related deep generative models in Table 1. Likelihood-based deep latent variable models based on the variational autoencoder (VAE) framework (Kingma & Welling, 2013)

have recently demonstrated impressive scene generation capabilities for high-resolution images with spatial latent variables split across a hierarchy of stochastic layers (Sønderby et al., 2016; Kingma et al., 2016; Child, 2020; Vahdat & Kautz, 2020). Differently, LOMs model perceptual data using scene and slot latent variables. During inference and generation, each slot is bound to a single entity in the scene which improves interpretability and compositionality (Greff et al., 2020).

Generative Neurosymbolic Machines (GNMs) (Jiang & Ahn, 2020) present the idea of combining a scene-level prior and symbolic latent variables within a hierarchical VAE for multi-object scene generation. GSGN (Deng et al., 2021) explores learning part-whole hierarchies for VAEs with similar symbolic latent variables. However, these VAEs make a strong assumption that the symbolic object-centric latent variables are *spatial* variables, i.e., bound to each cell of a fixed grid (Eslami et al., 2016; Crawford & Pineau, 2019; Lin et al., 2020). While this works well for synthetic image datasets with similarly sized objects, LOMs tackle the more challenging problem of learning a *scene-slot* hierarchy. LOMs fall into the category of VAEs that represent scenes as mixture models whose components are generated by a set of slots (Burgess et al., 2019; Greff et al., 2019; Engelcke et al., 2019; 2021; Locatello et al., 2020; Nanbo et al., 2020; Emami et al., 2021). EfficientMORL (EMORL) is an orderless slot-based VAE with a hierarchical independent slot prior that combines efficient bottom-up stochastic slot attention with top-down iterative refinement. GENESIS (GEN) (Engelcke et al., 2019) and GENESIS-v2 (GENv2) (Engelcke et al., 2021) use an autoregressive slot prior for multi-object scene generation. Unlike LOMs, neither possess a scene-level representation. Furthermore, while theoretically an autoregressive slot prior can learn dependencies between objects necessary for generating structured scenes, learning this distribution is difficult. GEN uses sequential attention to estimate an autoregressive posterior that imposes an unnatural order on objects, biases slot gradients, and does not work well on visually complex scenes (Engelcke et al., 2021). GENv2 replaces GEN's posterior with an orderless slot posterior which does not suffer from these limitations. However, orderless posteriors assume slots are independent which means GENv2's autoregressive prior will struggle to learn object dependencies.

Other VAEs with structured latent variables have been proposed for hierarchical scene generation (Anciukevicius et al., 2020; von Kügelgen et al., 2020) but make simplifying independence assumptions in their graphical models that prevents them from scaling to more complex scenes. The SetVAE (Kim et al., 2021) learns to generate set-structured data such as point clouds with a tree-structured hierarchy of sets of latent variables. However, unlike slot-based models, its latent variables are not explicitly trained to segregate the scene representation.

Generative adversarial methods for compositional scene synthesis (van Steenkiste et al., 2020; Nguyen-Phuoc et al., 2020; Liao et al., 2020; Ehrhardt et al., 2020; Niemeyer & Geiger, 2021; Hudson & Zitnick, 2021) avoid the computational challenges of Bayesian inference that LOMs address but lack the ability to *decompose* scenes into an interpretable hierarchy of representations.

## 3 LATENT OBJECT MODELS

### 3.1 GENERATIVE MODEL

We restrict the focus of this paper to unlabeled datasets $\mathcal{D}$ of i.i.d. samples of images $\mathbf{x} \in \mathbb{R}^{N \times C}$, $N = HW$, for clarity. Our goal is to fit the underlying data-generating process, which we propose to approach by using a three-level latent hierarchy. To that end, we define a scene-level latent variable $\mathbf{s} \in \mathbb{R}^{D_s}$, $K$ object-level latent variables $\mathbf{z}_1, \mathbf{z}_2, \ldots, \mathbf{z}_K, \mathbf{z}_k \in \mathbb{R}^{D_z}$ (or *slots*, which we abbreviate with $\mathbf{z}_{1:K}$), and pixel observations $\mathbf{x}_i \in \mathbb{R}^C$, $i = 1, \ldots, N$. The joint distribution over observed and latent variables can be factorized as follows:

$$p_\theta(\mathbf{x}, \mathbf{z}_{1:K}, \mathbf{s}) = p(\mathbf{s})p_\theta(\mathbf{z}_{1:K} \mid \mathbf{s})p_\theta(\mathbf{x} \mid \mathbf{z}_{1:K}). \tag{1}$$

We adopt the VAE framework for approximate Bayesian inference (Kingma & Welling, 2013; Rezende et al., 2014) since posterior inference is intractable and since we would like to parameterize these distributions with deep neural networks.

The scene-level prior $p(\mathbf{s})$ is left as a standard Gaussian in this work for simplicity. However, the slot prior is more complicated, since generating coherent scenes requires modeling relational information between slots. Unlike topic models, we cannot assume slots are i.i.d. when conditioned on $\mathbf{s}$. We

instead assume a *fixed and deterministic order* on the slots informed by $\mathbf{s}$:

$$p_\theta(\mathbf{z}_{1:K} \mid \mathbf{s}) = \prod_{k=1}^{K} p_\theta(\mathbf{z}_k \mid \boldsymbol{h}_k), \text{ where } \boldsymbol{h}_k = f_\theta(\boldsymbol{h}_{1:k-1}, \boldsymbol{s}), \tag{2}$$

where $f_\theta$ is a deterministic function that transforms a sample $\boldsymbol{s}$ from the scene prior and $\boldsymbol{h}_1 = f_\theta(\boldsymbol{s})$. Each $\boldsymbol{h}_k$ contains the order-aware information about which object the $k$th slot should generate. By designing the slot prior such that the $k$th slot indirectly depends on slots $\mathbf{z}_{1:k-1}$ through $\boldsymbol{h}_k$, the joint distribution over slots factorizes as a product of Gaussian conditionally independent (but not identically distributed) distributions. Notice that if we replaced each $\boldsymbol{h}_k$ with $\boldsymbol{s}$ the LOM generative model becomes a bag-of-words model.

The choice of image-likelihood strongly influences the complexity of the learned scene representation. We use a slot order-invariant Gaussian likelihood with the mean given by a sum of the product of the image-shaped decoded masks $m_k$ and components $x_k$:

$$p_\theta(\mathbf{x} \mid \mathbf{z}_{1:K}) = \prod_{i=1}^{N} \mathcal{N} \left( \sum_{k=1}^{K} m_{i,k} x_{i,k}, \sigma^2 \right). \tag{3}$$

The Gaussian likelihood is well-suited for generating complex scenes since it does not penalize solutions that use multiple slots to fit parts of the background (Locatello et al., 2020; Emami et al., 2021). By contrast, the Mixture-of-Gaussians likelihood used by some slot-based models (Burgess et al., 2019; Greff et al., 2019; Engelcke et al., 2019; 2021) heavily penalizes the model for splitting parts of the scene across multiple slots. A scene can be sampled from a LOM as follows:

1. Sample $\mathbf{s}$ from the scene prior $p(\mathbf{s})$.
2. Compute $\boldsymbol{h}_1 = f_\theta(\mathbf{s}), \boldsymbol{h}_2 = f_\theta(\boldsymbol{h}_1, \mathbf{s}), \ldots, \boldsymbol{h}_K = f_\theta(\boldsymbol{h}_{1:K-1}, \mathbf{s})$.
3. Sample one slot from each of the $p_\theta(\mathbf{z}_1 \mid \boldsymbol{h}_1), p_\theta(\mathbf{z}_2 \mid \boldsymbol{h}_2), \ldots, p_\theta(\mathbf{z}_K \mid \boldsymbol{h}_k)$.
4. Render the image by decoding the $K$ slots in parallel into masks $m_k$ and pixel components $x_k$ and then aggregating them with Equation 3.

## 3.2 INFERENCE

We have introduced how LOMs model the scene generation process. Now, we turn our attention to the inference algorithm for the slot and scene latent variables. We want to avoid directly estimating an ordered slot posterior using sequential scene decomposition since an optimal object ordering is unknown and difficult to learn without supervision. Our inference algorithm, visualized in Figure 2, uses an *auxiliary* generative model that assumes slots are orderless to indirectly obtain an ordered slot posterior. Orderless scene decomposition uses randomized iterative assignment to bind percepts to the slot variables (Greff et al., 2019; Locatello et al., 2020; Emami et al., 2021; Engelcke et al., 2021). The main idea behind our inference algorithm is to impose an ordering on the auxiliary orderless posterior with object ordering information provided by the deterministic variables $\boldsymbol{h}_{1:K}$. The algorithm shares similarities with recent work that proposes to learn a good canonical ordering for orderless sets to ease certain modeling tasks (Vinyals et al., 2016; Zhang et al., 2019). An important consequence of having an orderless slot posterior is that we can define a slot-order-invariant scene-level posterior:

$$q_\phi(\mathbf{s} \mid g_\phi(\mathbf{o}_{1:K})), \tag{4}$$

where $g_\phi$ is a permutation-invariant function. Orderless slots are denoted by $\mathbf{o}_{1:K}$. Each step of LOM inference is summarized here and visualized in Figure 2:

1. **Estimate orderless slot posterior:** Compute and sample from the orderless posterior $q_\omega$ estimated by the auxiliary model (with inference network parameters $\omega$): $\mathbf{o}_{1:K} \sim q_\omega(\mathbf{o}_{1:K} \mid \mathbf{x}) = \prod_{k=1}^{K} q_\omega(\mathbf{o}_k \mid \mathbf{x})$.
2. **Estimate scene posterior:** Compute and sample from the scene-level posterior: $\mathbf{s} \sim q_\phi(s \mid g_\phi(\mathbf{o}_{1:K}))$.
3. **Estimate scene-conditioned ordered slot prior:** Compute the ordered prior $p_\theta(\mathbf{z}_{1:K} \mid \mathbf{s})$ using a single sample $\mathbf{s}$ and the causal function $f_\theta$.

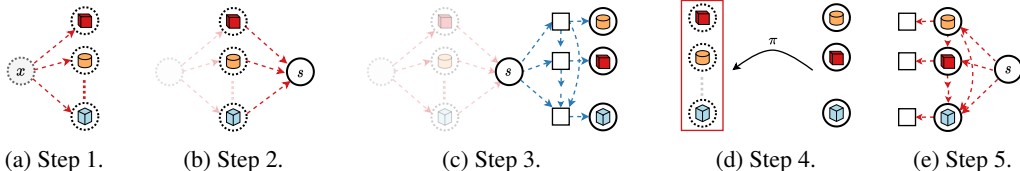

| (a) Step 1. | (b) Step 2. | (c) Step 3. | (d) Step 4. | (e) Step 5. |

Figure 2: Inference for Latent Object Models. Circles are slot latents. Dotted edges signify variables from the auxiliary orderless model and solid edges are variables from the LOM model. Boxes are deterministic variables. In steps 1-2 we infer an orderless slot posterior and the scene posterior. In step 3 we use the scene posterior to generate slots via the deterministic variables. In Step 4 we align the orderless slot posterior to the prior. Finally in step 5 we estimate new order-aware scale parameters for the aligned slot posterior through another set of deterministic variables.

4. **Align the orderless slot posterior:** Compute a matching $\pi$ between the $K$ means from $p_\theta(\mathbf{z}_{1:K} \mid \mathbf{s})$ and the $K$ means from $q_\omega(\mathbf{o}_{1:K} \mid \mathbf{x})$, e.g., by minimizing the pairwise $L2$ distance, then permute the order of the slot posterior means by $\pi$ to get $\mathbf{o}_{\pi(1):\pi(K)}$.

5. **Estimate an ordered slot posterior:** Compute new order-aware scale parameters $\sigma^{\mathbf{z}}_{\pi(k)}$ for the permuted slot posteriors to obtain *ordered* slot posteriors $q_\phi(\mathbf{z}_{\pi(k)} \mid \mathbf{h}'_{\pi(k)}, \mathbf{x})$.

Here $\sigma^{\mathbf{z}}_{\pi(k)} = w_\phi(\mathbf{h}'_{\pi(k)})$, with $\mathbf{h}'_{\pi(k)} = f'_\phi(\mathbf{o}_{\pi(1):\pi(k-1)}, \mathbf{s})$ and $w_\phi$ is a linear transformation. We introduce another causal function $f'_\phi$ to transform a sample from the scene posterior and the permuted orderless slots $\mathbf{o}_{\pi(1):\pi(K)}$ into the sequence $\mathbf{h}'_{\pi(1):\pi(K)}$. A simple and robust choice for computing $\pi$ is an $O(K)$ greedy alignment (Algorithm 1 in the appendix) that sequentially finds the best match for each slot. Each $\mathbf{h}'_{\pi(k)}$ provides the ordered slot posterior with its order-aware scale parameter:

$$\prod_{k=1}^{K} q_\phi(\mathbf{z}_{\pi(k)} \mid \mathbf{h}'_{\pi(k)}, \mathbf{x}) = \prod_{k=1}^{K} \mathcal{N}(\mu^{\mathbf{o}}_{\pi(k)}, (\sigma^{\mathbf{z}}_{\pi(k)})^2 I). \tag{5}$$

Here, $\mu^{\mathbf{o}}_{\pi(k)}$ is just the $k$th mean of the permuted orderless slot posterior.

**Architectures for $f_\theta, f'_\phi$ and $g_\phi$:** In this work we use a Transformer (Vaswani et al., 2017) with multiple blocks of single-head *causal* self-attention (SA) to implement $f_\theta$. The first token in the input is a sample $\mathbf{s} \sim p(\mathbf{s})$ and the next $K$ tokens are a shared trainable embedding initialized with zeros. We implement $f'_\phi$ using nearly the same causal Transformer architecture as $f_\theta$. The differences are that the first token is now a sample $\mathbf{s}$ from the scene posterior and instead of using a trainable embedding as input for the other tokens we pass the $K$ permuted orderless slots $\mathbf{o}_{\pi(1):\pi(K)}$. For $g_\phi$ we use a permutation-invariant Transformer with multiple blocks of residual SA and no positional encoding or causal masking. We aggregate the set output with a global sum pooling layer, which we find essential for capturing complex object relations in scenes. Details are provided in the appendix.

## 3.3 TRAINING

LOMs can be seen as a hybrid model that jointly trains an orderless VAE and an ordered VAE. Gradients are prevented from flowing back to the orderless VAE from the ordered VAE during training (via `stop_grad(o`$_{1:K}$`)`) to ensure that orderless scene decomposition remains unbiased. To estimate $q_\omega(\mathbf{o}_{1:K} \mid \mathbf{x})$, we can use any slot-based VAE with orderless decomposition. In this work, we train LOMs with both EMORL and GENv2. For simplicity, we re-use their respective decoder architectures to implement the LOM image likelihood.

**Objective:** The ELBO for training LOMs is maximized jointly with the ELBO for the auxiliary orderless generative model. The LOM ELBO resembles the ELBO used by previous deep generative models with a similar topic-model-inspired hierarchical factorization (Edwards & Storkey, 2016). We show it here presented as a sum of three terms and defer its derivation to the appendix.

First is a log-likelihood (i.e., reconstruction) term:

$$\mathcal{L}_{\text{LL}} = \mathbb{E}_{q_\phi(\mathbf{s}|g_\phi(\mathbf{o}_{1:K}))} \left[ \mathbb{E}_{q_\phi(\mathbf{z}_{\pi(1):\pi(K)}|\mathbf{h}'_{\pi(1):\pi(K)}, \mathbf{x})} \left[ \log p_\theta(\mathbf{x} \mid \mathbf{z}_{1:K}) \right] \right]. \tag{6}$$

Then, we have a reverse Kullback-Leibler (KL) term for the slots, which can be factorized as a sum over $K$ since the slot posterior and prior are conditionally independent and aligned via $\pi$:

$$\mathcal{L}_{\text{slotKL}} = -\mathbb{E}_{q_\phi(\mathbf{s}|g_\phi(\mathbf{o}_{1:K}))} \left[ \sum_{k=1}^{K} D_{KL}\big(q_\phi(\mathbf{z}_{\pi(k)} \mid \boldsymbol{h}'_{\pi(k)}, \mathbf{x}) \parallel p_\theta(\mathbf{z}_k \mid \boldsymbol{h}_k)\big) \right]. \tag{7}$$

Finally, we have a scene-level reverse KL term:

$$\mathcal{L}_{\text{sceneKL}} = -D_{KL}\big(q_\phi(\mathbf{s} \mid g_\phi(\mathbf{o}_{1:K})) \parallel p(\mathbf{s})\big). \tag{8}$$

Altogether, we maximize $\mathcal{L}$, the sum of these three terms and the auxiliary orderless model's ELBO:

$$\mathcal{L} = \mathcal{L}_{\text{orderless}} + \mathcal{L}_{\text{LL}} + \mathcal{L}_{\text{slotKL}} + \mathcal{L}_{\text{sceneKL}}. \tag{9}$$

**Impact of the ELBO on $f_\theta$:** Intuitively, maximizing the ELBO encourages $f_\theta$ to generate objects in a *consistent* order that results in a low average value for $\mathcal{L}_{\text{slotKL}}$. In practice, $f_\theta$ often learns to generate parts (or all) of the background with the same slot(s) and generate foreground objects using the same subset of slots (Figure 3).

**Optimization:** To dynamically balance the reconstruction and KL terms for all models we use GECO (Rezende & Viola, 2018). GECO reformulates maximization of the ELBO into a constrained minimization of the KL terms, where the constraint dictates a desired reconstruction quality $C$. Please refer to the appendix for a discussion on choosing $C$ as it is crucial for obtaining good performance. All other implementation details including architecture description, hyperparameters, and compute requirements are also deferred to the appendix.

## 4 EXPERIMENTS

**Datasets:** We use three synthetic multi-object datasets: *Objects Room* and *CLEVR6* from the standard Multi-Object Dataset (Kabra et al., 2019) and *ShapeStacks* (Groth et al., 2018). All three datasets consist of rendered images of 3D scenes containing 3+ objects and have variable illumination, shadow, and camera perspectives. Successful object-centric scene generation requires reasoning about local object relations (e.g., occlusion in CLEVR6, placement of objects relative to the floor in Objects Room) as well as global relations (e.g., stacking multiple blocks in ShapeStacks). We focus on synthetic scenes in this work since we wish to first establish correctness of the proposed LOM framework before tackling real-world scenes.

**Baselines:** We compare LOM-EMORL and LOM-GENv2-G against the state-of-the-art slot-based VAE for scene generation GENv2, as well as EMORL, GNM, GENESIS (GEN), and the NVAE. EMORL has an independent slot prior that does not capture dependencies between objects; therefore, it serves as a simple scene generation baseline. We train GENv2 with the default Mixture-of-Gaussians image likelihood (GENv2-MoG) on CLEVR6 as these results were not reported in their paper. Results for GEN and GENv2-MoG on the other two datasets are provided by Engelcke et al. (2021). We train GENv2 with the Gaussian likelihood on all three datasets (GENv2-G) for a more direct comparison to the LOMs. See Section 2 for a detailed discussion on these models and the appendix for training details and hyperparameters.

**Metrics:** We use multiple metrics to analyze the performance of all models in terms of object-centric decomposition and disentanglement, image generation quality, and the structural accuracy of generated scenes. The Frechet Inception Distance (**FID**) (Heusel et al., 2017) is a standard metric for quantifying the ability of a model to generate images similar to the training set. We follow Engelcke et al. (2021) and use 10K real and generated samples to calculate this score. However, we found that the FID score heavily penalizes blurriness in generated images while ignoring whether the semantics of the generated scenes are accurate. Therefore, inspired by the evaluation of GNM (Jiang & Ahn, 2020), on ShapeStacks we generate 100 images with each model and count the fraction of images containing a stack where all blocks in the stack are touching each other (structural accuracy, or **S-Acc**). To compare the ability to jointly generate high-quality scenes and learn object-centric representations, we compute the adjusted rand index for segmented foreground objects (**ARI-FG**) (Rand, 1971; Hubert & Arabie, 1985) on all datasets and disentanglement scores (**DCI**) (Eastwood & Williams, 2018) on CLEVR6. The NVAE is only trained on CLEVR6 since it is the only considered dataset that provides ground-truth latent factors.

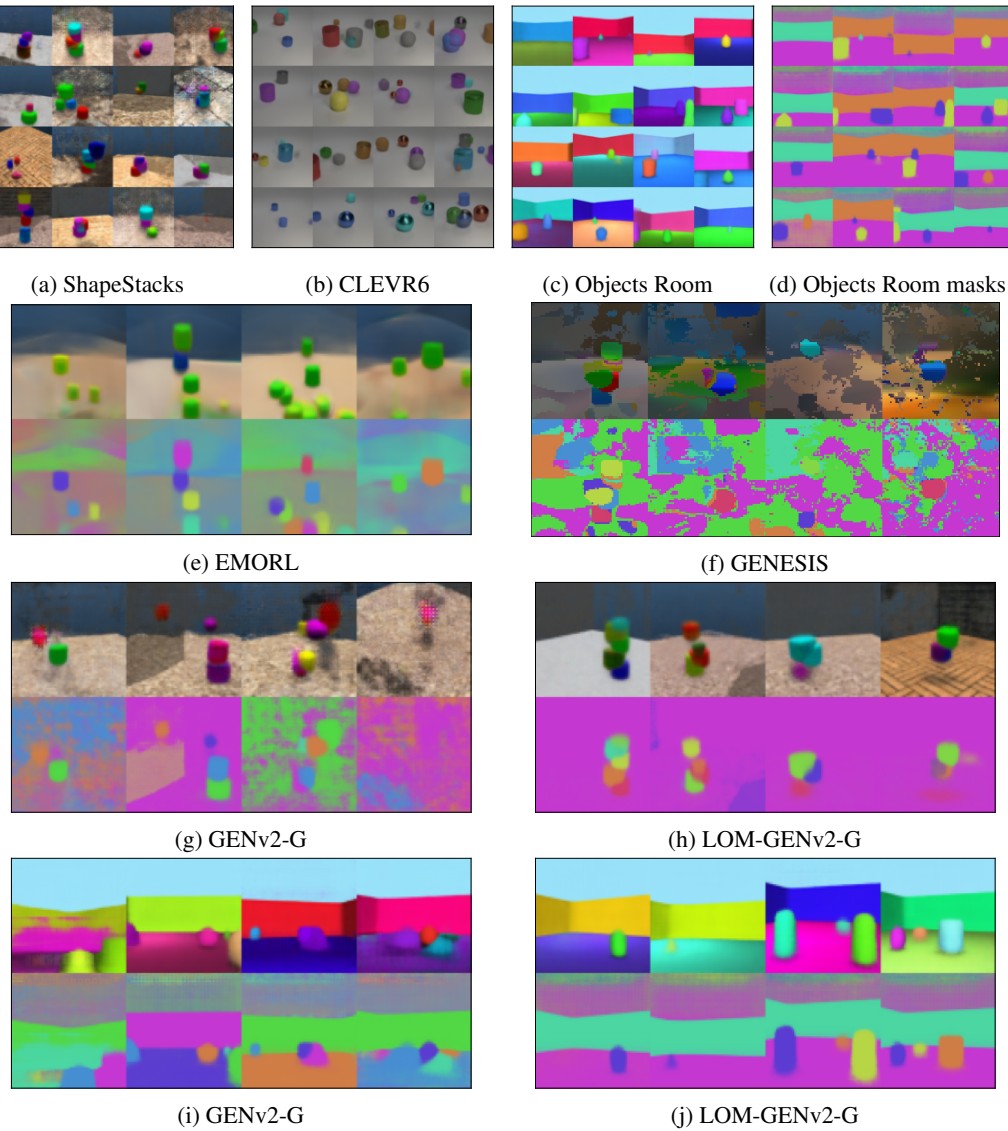

(a) ShapeStacks      (b) CLEVR6      (c) Objects Room      (d) Objects Room masks

(e) EMORL                (f) GENESIS

(g) GENv2-G                (h) LOM-GENv2-G

(i) GENv2-G                (j) LOM-GENv2-G

Figure 3: (a-c) Unconditional samples from LOM (GenV2-G). d) LOMs learn a relatively stable and deterministic assignment of objects to specific slots for generating scenes. The color of each segment indicates the slot number. e-h) A few generated ShapeStacks scenes. GEN scenes are generated with a trained model released by the authors. LOM (GENv2-G) generates scenes that most accurately reflect multi-object structure in the data and with the fewest visual artifacts.

We ablate attributes of LOMs on a 48x48 resolution variant of Objects Room but defer these results to the appendix (Section A.5). Specifically, we measure the importance of: the expressivity of $f_\theta, f'_\theta$, and $g_\phi$, computing order-aware scale parameters during LOM inference, and *not* sharing parameters between $f_\theta$ and $f'_\theta$. We also consider two variants with autoregressive slot priors where we try to (unsuccessfully) train them by directly minimizing the KL with respect to an orderless slot posterior.

## 4.1 QUALITATIVE EVALUATION

We first verify that LOMs are able to fit the data-generating process for each dataset by inspecting unconditional samples from the prior (Figures 3a-3d). Note that we slightly scale down the standard deviations in the prior using a temperature of $\tau = 0.8$ when sampling, which is common practice for hierarchical VAEs (Vahdat & Kautz, 2020) to help keep samples in high probability regions. See

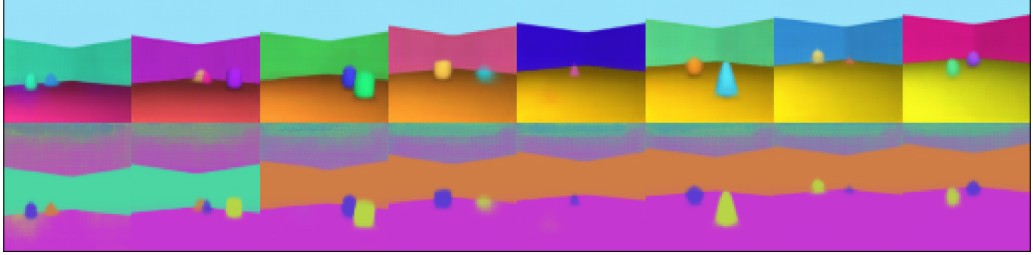

Figure 4: **LOMs can generate interpolated scenes between two random samples from the scene prior by smoothly changing individual object and scene-level attributes**. We use spherical linear interpolation to generate the intermediate latents between two random samples from $p(\mathbf{s})$. This suggests the LOM has learned a good scene-level manifold.

Figure 7 in the appendix for a temperature sensitivity analysis. Upon close inspection of the Objects Room scene masks (Figures 3d and 3j), we see that LOMs can consistently generate the wall, floor, and shapes with the same groups of slots (as indicated by mask color). By learning a consistent and deterministic order with which to generate objects, LOMs are able to learn important dependencies between objects needed for generating scenes.

We compare Objects Room and ShapeStacks samples from slot-based VAEs in Figures 3e-3j, which are the two datasets with the most complex multi-object structure. As expected, EMORL generates incoherent scenes with floating blocks due to its independent prior. GEN generates seemingly incoherent scenes from the visually challenging ShapeStacks dataset. In theory, its autoregressive posterior should be capable of learning ordered dependencies between slots; however, GEN inference tends to heavily rely on color cues, which fails in datasets with textures and shadows. Samples from GENv2-G's autoregressive prior contain visual artifacts and structural inaccuracies (e.g., extra walls, missing floors in Objects Room, floating blocks in ShapeStacks). Its orderless posterior cannot capture object dependencies, which makes it difficult for the autoregressive prior to learn to sequentially generative scenes from the challenging Objects Room and ShapeStacks datasets. By contrast, we see that LOMs use their ability to learn a deterministic generation order and their scene-level representation to generate scenes that more accurately reflect the ground truth data.

It is also helpful to inspect the LOM scene-level latent space by interpolating between two random samples from the scene prior using spherical linear interpolation (Figure 4). The interpolated images resemble the training data which suggests that the LOM has learned a good scene-level manifold. Scene-level attributes such as camera perspective and floor color change smoothly and the number of objects roughly remains at two. More interpolation examples are provided in the appendix (Figures 10, 11, 12) as well as qualitative comparisons (Figures 13-15). Figures 16-20 visualize reconstructions and segmentations for all models. We also explore how LOMs can hierarchically cluster a scene dataset using scene-level and object-level relationships in the appendix (Section A.6).

## 4.2 QUANTITATIVE EVALUATION

We present the results in Table 2. GNM has difficulty discovering and segmenting individual objects in Objects Room and ShapeStacks which leads to much lower ARI-FG scores compared to the slot-based models (see, e.g., Figures 14 and 19). Despite this, GNM achieves low FID scores on all three environments. Its inability to decompose scenes in two out of three environments suggests that the slot-based approaches are more promising for likelihood-based multi-object scene generation.

Out of the slot-based models, LOM-GENv2-G achieves the best FID score and S-Acc on ShapeStacks. LOM-EMORL achieves the best FID score on CLEVR6. To quantify the improvement in generation quality gained by the LOM framework, we can compare FID and S-Acc scores for EMORL with LOM-EMORL and GENv2-G with LOM-GENv2-G. Removing sum pooling in the scene encoder $g_\phi$ decreases LOM-EMORL's S-Acc by a 36%, highlighting the role it plays in encoding relations

Table 2: Mean $\pm$ std. dev. over three training runs. n.s.p. = no sum pooling.

| Model | CLEVR6 | | Objects Room | | ShapeStacks | | |
|---|---|---|---|---|---|---|---|
| | FID↓ | ARI-FG↑ | FID↓ | ARI-FG↑ | FID↓ | ARI-FG↑ | S-Acc (%)↑ |
| GNM | $27.5_{\pm 1}$ | $0.97_{\pm 0.01}$ | $51.6_{\pm 5}$ | $0.23_{\pm 0.05}$ | $49.3_{\pm 2}$ | $0.37_{\pm 0.07}$ | 100 |
| GEN | - | - | $62.8_{\pm 3}$ | $0.63_{\pm 0.03}$ | $186.8_{\pm 18}$ | $0.70_{\pm 0.05}$ | 0 |
| GENv2-MoG | $61.0_{\pm 3}$ | $0.98_{\pm 0.00}$ | $\mathbf{52.6}_{\pm 3}$ | $\mathbf{0.84}_{\pm 0.01}$ | $112.7_{\pm 3}$ | $\mathbf{0.81}_{\pm 0.00}$ | 59 |
| EMORL | $244.0_{\pm 19}$ | $0.96_{\pm 0.02}$ | $178.3_{\pm 27}$ | $0.47^{\dagger}_{\pm 0.22}$ | $258.4_{\pm 57}$ | $0.60_{\pm 0.04}$ | 0 |
| LOM-EMORL | $\mathbf{52.0}_{\pm 6}$ | $0.96_{\pm 0.01}$ | $67.4_{\pm 3}$ | $0.51^{\dagger}_{\pm 0.25}$ | $132.2_{\pm 15}$ | $0.60_{\pm 0.04}$ | 76 |
| LOM-EMORL-n.s.p. | - | - | - | - | $139.1_{\pm 10}$ | $0.62_{\pm 0.08}$ | 40 |
| GENv2-G | $61.0_{\pm 3}$ | $0.98_{\pm 0.00}$ | $87.6_{\pm 4}$ | $0.80_{\pm 0.00}$ | $115.3_{\pm 6}$ | $0.68_{\pm 0.02}$ | 42 |
| LOM-GENv2-G | $62.3_{\pm 8}$ | $0.96_{\pm 0.00}$ | $63.1_{\pm 8}$ | $0.79_{\pm 0.00}$ | $\mathbf{91.6}_{\pm 3}$ | $0.68_{\pm 0.02}$ | $\mathbf{82}$ |

† The 3 ARI-FG scores are EMORL: 0.60, 0.64, 0.16 and LOM (EMORL): 0.69, 0.70, 0.16.

Table 3: Comparing object-centric representation quality of models with spatial latents (NVAE) and scene-slot hierarchical latents on CLEVR6. Metrics are mean $\pm$ std. dev. over three training runs.

| Model | FID↓ | ARI-FG↑ | Disentanglement↑ | Completeness↑ | Informativeness↑ |
|---|---|---|---|---|---|
| LOM-EMORL | $52.0_{\pm 6}$ | $\mathbf{0.96}_{\pm 0.01}$ | $\mathbf{0.48}_{\pm 0.04}$ | $\mathbf{0.45}_{\pm 0.06}$ | $\mathbf{0.25}_{\pm 0.03}$ |
| NVAE | $\mathbf{28.3}_{\pm 4}$ | - | $0.19_{\pm 0.0}$ | $0.16_{\pm 0.0}$ | $0.0_{\pm 0.0}$ |

involving many objects. In almost all settings, we observe clear improvements. LOM-GENv2-G on CLEVR6 did not improve the FID compared to GENv2-G despite there being a noticeable qualitative improvement (Figure 15). We believe this is because GENv2's encoder introduces visual artifacts that artificially inflates the FID score, but this requires further investigation.

GENv2-MoG achieves lower FID scores than GENv2-G in Objects Room. We observed that GENv2-MoG segments the walls, floor, and sky into a single slot (Figure 13), whereas GENv2-G tends to segment the wall and floor separately (Figure 3i). GENv2's orderless posterior's inability to learn slot dependencies means that GENv2-G has the more difficult task of orienting the objects, walls, and floor during generation whereas GENv2-MoG simplifies this task, leading to GENv2-G's worse FID scores. By contrast, LOM-GENv2-G successfully learns dependencies between objects and clearly improves GENv2-G's Objects Room FID score (Figure 3j). We believe that GEN achieves a lower FID score than GENv2-G in Objects Room because GEN's autoregressive posterior, which is heavily reliant on color cues, is able to successfully model slot dependencies this dataset. GEN clearly has difficulty with the more challenging ShapeStacks dataset and achieves much higher FID scores. In general, LOMs maintain the decomposition performance (as measured by ARI-FG) as the auxiliary orderless VAE. Finally, Table 3 shows that although the NVAE achieves low FID on CLEVR6, its poor disentanglement scores suggests that its spatial latents do not encode interpretable object-centric information about scenes.

## 5 CONCLUSION

We presented a probabilistic framework for learning an interpretable hierarchy of scene and slot latent variables. The learned representation shares advantages of both a single scene-level representation as well as object-centric slots. This work improves our understanding on how to develop object-centric VAEs, particularly with respect to accurately modeling scene-level statistics.

We believe the interpretation of the deterministic variables in the LOM graphical model could benefit from further study. They are essential to the success of LOMs but trade off the ability to model stochasticity in the order in which objects are generated. We observed that LOMs occasionally generate objects that intersect each other due to a lack of 3D understanding in the latent representations. Although unsupervised 3D scene inference is particularly difficult due to ambiguities introduced by perspective geometry, obtaining 3D object representations should improve generation quality (Chen et al., 2020). Our LOM implementation could also be improved by replacing the standard Gaussian scene prior with a more flexible distribution such as a mixture-of-Gaussians.

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

# A  APPENDIX

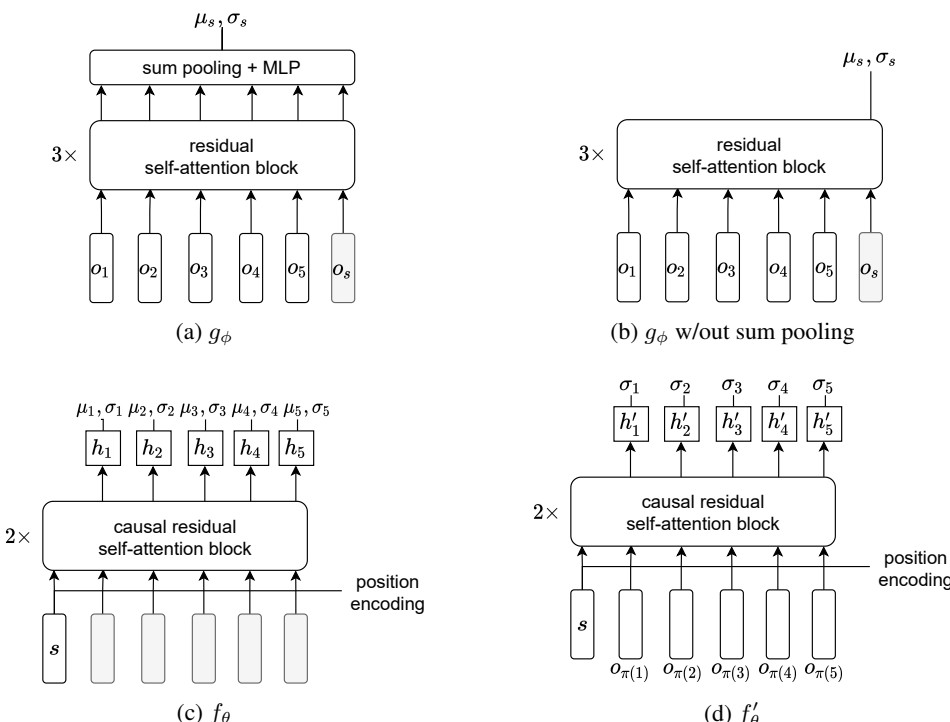

Figure 5: The Transformer architectures used to implement the LOM. We visualize each with $K = 5$. The output(s) of each function is mapped to Gaussian parameters $\mu, \sigma$, which we annotate onto each diagram. a) The scene encoder $g_\phi$. We do not use positional encoding or causal attention masking in the slot-order-invariant scene encoder. The last input $\mathbf{o}_s$ is a trainable embedding. b) For the ablation study on the impact of the global sum pooling layer on $g_\phi$, we directly map the $(K + 1)^{\text{st}}$ element of the output set to the Gaussian scene posterior parameters. c) The function $f_\theta$ estimates the ordered slot prior parameters from a sample from the scene posterior $\mathbf{s}$ and a shared trainable embedding (grey). d) We estimate the ordered slot posterior scale parameters with the outputs $\mathbf{h}'$ of $f'_\theta$.

## A.1  NEURAL ARCHITECTURES

We describe the details of the neural architectures used in this paper to implement LOMs. We use GELU non-linear activations (Hendrycks & Gimpel, 2016) and xavier weight initialization (Glorot & Bengio, 2010).

**Residual self-attention block:** We construct the permutation-invariant scene encoder and the causal slot prior/posterior by stacking residual self-attention (SA) blocks (Figure 5). Each SA block is based on the Transformer (Vaswani et al., 2017) design. For each such block, we apply single-head scaled dot-product attention to an input $\boldsymbol{x} \in \mathbb{R}^{N \times D}$:

$$\alpha = \texttt{softmax}(\frac{k(\boldsymbol{x})q(\boldsymbol{x})^{\mathsf{T}}}{\sqrt{D}})v(\boldsymbol{x}) \tag{10}$$

$$\boldsymbol{x} = \boldsymbol{x} + \texttt{MLP}(\texttt{LayerNorm}(\boldsymbol{x} + \alpha)). \tag{11}$$

The $k, q, v$ linear maps are $D \times D_z$ projections and the MLP is a $D_z \times D_z$ projection. In our experiments we set $D_z$ to 64.

**Scene encoder $g_\phi$:** The scene encoder $g_\phi$ (Figure 5a) has 3 residual SA blocks. It takes as input an orderless set of slots $\mathbf{o}_{1:K}$ where each slot is a $D_z$-dim vector. We use the means of the orderless slot posterior for the input in practice. We append a $D_z$-dim trainable scene embedding $\mathbf{o}_s$ at position

---

**Algorithm 1** Greedy matching algorithm for $\boldsymbol{\pi}$. Uses $O(K)$ time and $O(K^2)$ space.

---

1: **Input:** Ordered means $\mu_1^{\mathbf{z}}, \ldots, \mu_K^{\mathbf{z}}$, orderless means $\mu_1^{\mathbf{o}}, \ldots, \mu_K^{\mathbf{o}}$, with $\mu_k \in \mathbb{R}^{D_z}$.
2: $\boldsymbol{C}[i, j] = \|\mu_i^{\mathbf{z}} - \mu_j^{\mathbf{o}}\|_2, \ \forall(i, j)$
3: $\pi = [\,]$
4: **for** index $i = 1 \ldots K$ **do**
5: $\quad j^* = \arg\min \boldsymbol{C}[i, :]$
6: $\quad \boldsymbol{C}[:, j^*] = +\inf$
7: $\quad \pi[i] = j^*$
8: **end for**
9: **return** Permutation $\pi$

---

$K + 1$ so that the input to $g_\phi$ is a set of size $K + 1$. After the 3 residual SA blocks we apply a sum pooling layer that aggregates the output set by summing the $K + 1$ vectors and projecting the result to the scene latent dimension $D_s$ with an MLP. Then, two linear layers are used to obtain the mean and softplus-activated standard deviation for the Gaussian scene posterior. Our ablation study analyzes the number of residual SA blocks and the impact of the sum pooling layer on the ability to capture higher-order object relations (Figure 5b).

**Causal Transformer** $f_\theta$**:** A straightforward architectural choice for implementing $f_\theta$ is a recurrent neural network (RNN) such as a GRU (Cho et al., 2014), which have a linear time dependency on the input sequence length. By contrast, Transformers have a quadratic dependency on the input sequence length; however, this computation can be efficiently parallelized, resulting in a constant time dependency in practice. Moreover, they have a superior ability to capture long-range interactions between elements of the sequence due to the use of self-attention (Radford et al., 2019; Brown et al., 2020; Zhou et al., 2021). Therefore we use a Transformer with single-head *causal* self-attention (SA) to implement $f_\theta$. In detail, the first token in the input sequence is the sample $\mathbf{s} \sim p(\mathbf{s})$ and the subsequent $K$ tokens in the sequence are set to a shared trainable embedding of dimension $D_z$ initialized with zeros. The scene sample is projected to the same dimension as the $D_z$-dimensional embeddings with a single layer MLP. To provide extra positional information we add a positional encoding to the input sequence. This is implemented by mapping the values $[0/(K + 1), 1/(K + 1), \ldots, (K + 1)/(K + 1)]$ to $D_z$-dimensional vectors with a linear layer and adding this to the input. We use the standard approach for implementing causal self-attention by masking out attention values in each of the residual SA blocks. The length $K$ output sequence, starting from the second output token, are the $\boldsymbol{h}_{1:K}$. These are each mapped to Gaussian parameters for each $p_\theta(\mathbf{z}_k \mid \boldsymbol{h}_k)$ with two linear layers. We train $f_\theta$ end-to-end alongside the inference and generation networks to maximize the ELBO, which we discussed in Section 3.3.

**Causal Transformer** $f_\theta'$**:** For the causal Transformer that predicts the order-dependent scale parameters for the ordered slot posterior, we use the same architecture as $f_\theta$ except for a few minor differences. Primarily, the inputs are now a sample from the scene *posterior* and the means of the permuted orderless slot posterior. The outputs of the stack of SA blocks are mapped with a single linear layer to the $D_z$-dim softplus-activated standard deviations of the ordered slot posterior. The means of the ordered slot posterior are fixed to be the means of the permuted orderless posterior.

## A.2 GREEDY MATCHING ALGORITHM

Pseudocode for the greedy matching algorithm we use to align the orderless slot posterior to the ordered slot prior is provided in Algorithm 1. Alternative matching algorithms such as the Hungarian algorithm (Munkres, 1957) can be used instead at a higher computational cost. We did not empirically explore other matching algorithms since the greedy solution worked well initially.

## A.3 DERIVATION OF THE LOM ELBO

We assume that we are given a sample $\mathbf{o}_{1:K}$ from an orderless slot posterior estimated by an auxiliary orderless generative model. We can derive a lower-bound on the log marginal likelihood as follows:

$$\log p(\mathbf{x}) = \log \int p_\theta(\mathbf{x}, \mathbf{s}, \mathbf{z}_{\pi(1):\pi(K)}) d\mathbf{s}, d\mathbf{z}_{1:K} \tag{12}$$

$$= \log \int \frac{q_\phi(\mathbf{s}, \mathbf{z}_{\pi(1):\pi(K)} \mid \mathbf{x})}{q_\phi(\mathbf{s}, \mathbf{z}_{\pi(1):\pi(K)} \mid \mathbf{x})} p_\theta(\mathbf{x}, \mathbf{s}, \mathbf{z}_{1:K}) d\mathbf{s}, d\mathbf{z}_{1:K} \tag{13}$$

$$= \log \mathbb{E}_q \left[ \frac{p_\theta(\mathbf{x}, \mathbf{s}, \mathbf{z}_{1:K})}{q_\phi(\mathbf{s}, \mathbf{z}_{\pi(1):\pi(K)} \mid \mathbf{x})} \right]. \tag{14}$$

Applying Jensen's inequality, we get:

$$\log p(\mathbf{x}) \geq \mathbb{E}_{q_\phi(\mathbf{s}, \mathbf{z}_{\pi(1):\pi(K)} \mid \mathbf{x})} \left[ \log \frac{p_\theta(\mathbf{x}, \mathbf{s}, \mathbf{z}_{1:K})}{q_\phi(\mathbf{s}, \mathbf{z}_{\pi(1):\pi(K)} \mid \mathbf{x})} \right] \tag{15}$$

$$= \mathbb{E}_{q_\phi(\mathbf{s} \mid g_\phi(\mathbf{o}_{1:K}))} \left[ \mathbb{E}_{q_\phi(\mathbf{z}_{\pi(1):\pi(K)} \mid \mathbf{h}'_{\pi(k)}, \mathbf{x})} \left[ \log \frac{p_\theta(\mathbf{x} \mid \mathbf{z}_{1:K}) p_\theta(\mathbf{z}_{1:K} \mid \mathbf{s}) p(\mathbf{s})}{q_\phi(\mathbf{s} \mid g_\phi(\mathbf{o}_{1:K})) q_\phi(\mathbf{z}_{\pi(1):\pi(K)} \mid \mathbf{h}'_{\pi(k)}, \mathbf{x})} \right] \right]. \tag{16}$$

We can treat the inner log product of terms as a sum of logs to get the following three loss terms. First is a log-likelihood loss term:

$$\mathcal{L}_{\text{LL}} = \mathbb{E}_{q_\phi(\mathbf{s} \mid g_\phi(\mathbf{o}_{1:K}))} \left[ \mathbb{E}_{q_\phi(\mathbf{z}_{\pi(1):\pi(K)} \mid \mathbf{h}'_{\pi(k)}, \mathbf{x})} \left[ \log p_\theta(\mathbf{x} \mid \mathbf{z}_{\pi(1):\pi(K)}) \right] \right]. \tag{17}$$

Second is a slot reverse KL divergence loss:

$$\mathcal{L}_{\text{KL\_z}} = \mathbb{E}_{q_\phi(\mathbf{s} \mid g_\phi(\mathbf{o}_{1:K}))} \left[ \mathbb{E}_{q_\phi(\mathbf{z}_{\pi(1):\pi(K)} \mid \mathbf{h}'_{\pi(k)} \mathbf{x})} \left[ \log \frac{p_\theta(\mathbf{z}_{1:K} \mid \mathbf{s})}{q_\phi(\mathbf{z}_{\pi(1):\pi(K)} \mid \mathbf{h}'_{\pi(k)}, \mathbf{x})} \right] \right] \tag{18}$$

$$= -\mathbb{E}_{q_\phi(\mathbf{s} \mid g_\phi(\mathbf{o}_{1:K}))} \left[ D_{KL} \big( q_\phi(\mathbf{z}_{\pi(1):\pi(K)} \mid \mathbf{h}'_{\pi(k)}, \mathbf{x}) \parallel p_\theta(\mathbf{z}_{1:K} \mid \mathbf{s}) \big) \right]. \tag{19}$$

Since both the ordered slot posterior and the ordered slot prior are products of $K$ conditionally independent distributions, this reverse KL further factorizes as a sum over $K$:

$$\mathcal{L}_{\text{KL\_z}} = -\mathbb{E}_{q_\phi(\mathbf{s} \mid g_\phi(\mathbf{o}_{1:K}))} \left[ \sum_{k=1}^K D_{KL} \big( q_\phi(\mathbf{z}_{\pi(k)} \mid \mathbf{h}'_{\pi(k)}, \mathbf{x}) \parallel p_\theta(\mathbf{z}_k \mid \mathbf{h}_k) \big) \right]. \tag{20}$$

Finally, there is a scene-level reverse KL divergence loss:

$$\mathcal{L}_{\text{KL\_s}} = \mathbb{E}_{q_\phi(\mathbf{s} \mid g_\phi(\mathbf{o}_{1:K}))} \left[ \mathbb{E}_{q_\phi(\mathbf{z}_{\pi(1):\pi(K)} \mid \mathbf{h}'_{\pi(k)}, \mathbf{x})} \left[ \log \frac{p(\mathbf{s})}{q_\phi(\mathbf{s} \mid g_\phi(\mathbf{o}_{1:K}))} \right] \right] \tag{21}$$

$$= \mathbb{E}_{q_\phi(\mathbf{s} \mid g_\phi(\mathbf{o}_{1:K}))} \left[ \log \frac{p(\mathbf{s})}{q_\phi(\mathbf{s} \mid g_\phi(\mathbf{o}_{1:K}))} \right] \tag{22}$$

$$= -\mathbb{E}_{q_\phi(\mathbf{s} \mid g_\phi(\mathbf{o}_{1:K}))} \left[ \log \frac{q_\phi(\mathbf{s} \mid g_\phi(\mathbf{o}_{1:K}))}{p(\mathbf{s})} \right] \tag{23}$$

$$= -D_{KL} \big( q_\phi(\mathbf{s} \mid g_\phi(\mathbf{o}_{1:K})) \parallel p(\mathbf{s}) \big). \tag{24}$$

Therefore, we have the following ELBO we wish to maximize with respect to parameters $\theta, \phi$:

$$\log p(\mathbf{x}) \geq \mathcal{L}_{\text{LL}} + \mathcal{L}_{\text{KL\_z}} + \mathcal{L}_{\text{KL\_s}}. \tag{25}$$

In practice, we approximate this ELBO by averaging over a minibatch of dataset samples of size $B$ and use stochastic gradient ascent.

## A.4 EXPERIMENT DETAILS

An open source implementation of the code for replicating all experiments will be released with link placed here. Videos demonstrating latent space random walks will also be released. We provide the generated samples and our labels used to compute the structure accuracy metric in the supplementary material.

### A.4.1 Hyperparameters

**Auxiliary orderless models:** We trained two LOM variants, one that uses EfficientMORL (Emami et al., 2021) (EMORL) to extract the auxiliary orderless posterior and one which uses GENESIS-v2 (GENv2) (Engelcke et al., 2021). The authors of both models have released open source implementations.[1][2] For EMORL, we keep their architecture and hyperparameters fixed except that the number of iterative refinement steps is set at 2 for all experiments. We use the Gaussian image likelihood with a global standard deviation of 0.7 for the Objects Room and ShapeStacks datasets. For CLEVR6, we needed to lower the global standard deviation to 0.1 to achieve sharp object segmentations. We use the refined approximate EMORL posterior for LOM inference. When using GENv2 to implement a LOM, we disable the autoregressive prior and replace it with an independent Gaussian slot prior. We also used a Gaussian image likelihood instead of the default mixture-of-Gaussians so that both LOM-EMORL and LOM-GENv2-G used the same likelihood. LOM-GENv2-G uses a global standard deviation of 0.7 for the Gaussian likelihood on all three environments. We use the approximate posterior estimated from the output of GENv2's non-parametric clustering algorithm for LOM inference. We set the slot dimension to 64 for all models. Please see the respective papers and code for EMORL and GENv2 for complete architecture and hyperparameter details.

**LOM:** We maintain the same slot dimension ($D_z = 64$) and image likelihoods as the auxiliary orderless models on all environments. The scene latent dim $D_s$ is 128 which is chosen to be twice the slot dimension. We use the same decoder architectures from EMORL and GENv2, respectively, for mapping slots to image-shaped masks and components. The optimization hyperparameters are also kept the same as EMORL and GENv2 for each LOM. For LOM-EMORL, both the auxiliary model and the LOM are trained jointly with the Adam optimizer (Kingma & Ba, 2014) with default PyTorch (Paszke et al., 2019) hyperparameters and a learning rate of 4e-4. EMORL's learning rate schedule is a linear warmup for 10K steps then multiplicative decay with by a rate of 0.5 every 100K steps. LOM-GENv2-G is trained in a similar manner, again using Adam and a learning rate of 1e-4 but without any learning rate schedule.

**Datasets:** Following Engelcke et al. (2021) we use $K = 7$ and $K = 9$ slots respectively for Objects Room and ShapeStacks. For CLEVR6, we use $K = 7$ slots following standard practice on this popular multi-object benchmark. LOM-EMORL pre-processes all images by converting RGB values to the range $[-1, 1]$ before passing them as input to the model. Objects Room can be accessed freely online.[3] This dataset contains 64x64 RGB training images. The ShapeStacks dataset is also freely available for download online and also has 64x64 RGB images.[4]. We use the same preprocessing protocol for CLEVR6 as Emami et al. (2021), which is to center crop the images to 192x192 and the resize them to size 96x96. This dataset is available as part of the Multi-Object Dataset.

### A.4.2 Balancing reconstruction and KL

**GECO:** Both EMORL and GENv2 use GECO to balance reconstruction and KL losses (Rezende & Viola, 2018). We use the respective implementations of GECO from each code base for training LOM-EMORL and LOM-GENv2-G.

For LOM-EMORL we customize the GECO target for each dataset, which is dependent on the image resolution and the choice of image likelihood. For Objects Room, we used a per-pixel and per-channel target of -0.352 (for an unnormalized Gaussian log-likelihood). Due to the increased visual complexity of ShapeStacks, we lowered this target to -0.351 to achieve a better trade-off of KL and reconstruction. On CLEVR6 we used a target of -2.265 (for an unnormalized Gaussian log-likelihood with $\sigma = 0.1$). The rule of thumb used to tune the GECO target is that the target should be reached after about 15-20% of the training steps. After the target is reached, the GECO Lagrange parameter is automatically decreased to 1 so that a valid ELBO is maximized. We use a GECO learning rate of 1e-6 for updating the Lagrange parameter and multiply this by an acceleration factor once the target is reached. Acceleration factors of 5, 10, and 50 were used on Objects Room, CLEVR6, and ShapeStacks respectively.

---

[1] https://github.com/pemami4911/EfficientMORL
[2] https://github.com/applied-ai-lab/genesis
[3] https://github.com/deepmind/multi_object_datasets
[4] https://ogroth.github.io/shapestacks/

As mentioned, LOM-GENv2-G is trained to optimize an unnormalized Gaussian image likelihood like LOM-EMORL. For Objects Room and ShapeStacks we used a per-pixel and per-channel target of -0.353 and increased this to -0.356 for CLEVR6. We found that we needed to decrease the GECO learning rate from 1e-5 to 1e-6 for these datasets compared to the default GECO learning rate used by GENv2. We note that the GENv2 codebase had a better GECO implementation than EMORL that was less sensitive to particular hyperparameters and with more robust annealing schedules. Improving the GECO implementation used by EMORL should help make tuning these hyperparameters easier for replication.

**Freebits:** We found it beneficial to use a *freebits* parameter $\lambda$ on some of the KL terms to help avoid posterior collapse (Maaløe et al., 2019). In detail, for a minibatch of size $B$ the freebits-augmented scene KL is:

$$D_{KL}\big(q_\phi(\mathbf{s} \mid g_\phi(\mathbf{o}_{1:K})) \parallel p(\mathbf{s})\big) \approx \frac{1}{B} \sum_{i=1}^{B} \max\big(\lambda, D_{KL}\big(q_\phi(\mathbf{s} \mid g_\phi(\mathbf{o}_{1:K}^i)) \parallel p(\mathbf{s})\big)\big). \quad (26)$$

This encourages the model to keep the KL at least as large as $\lambda$. For slot-based distributions that factorize as products over $K$, we take the max of $\lambda$ and the sum over $K$ slot-wise KL terms. We use this for each of EMORL's hierarchical KL terms as well as the LOM scene KL (but not for the LOM slot KL since the posterior means are fixed via the stopped gradients, meaning that we would like to push this KL as close to zero as possible). We use freebits for the LOM-GENv2-G scene KL as well. We use $\lambda = 0.75 \times D_z$ for each KL term in EMORL and $\lambda = 0.25 \times D_s$ for the LOM scene KL. Inspired by Maaløe et al. (2019) we tried annealing $\lambda$ down slightly to $0.167 \times D_s$ for the scene KL term during the last 25K steps of training and found that this slightly increased performance.

### A.4.3 BASELINES AND COMPUTE

**GENESIS and GENESIS-v2:** The authors of GENv2 have released pre-trained weights for GEN trained on ShapeStacks and GENv2 trained on Objects Room and ShapeStacks. We use these weights for model visualizations and to compute the structure accuracy metric. To train GENv2 on CLEVR6 with the default mixture-of-Gaussians image likelihood, we had to lower the standard deviation to 0.1 to get it to work decently well. We adjusted the GENv2 GECO target accordingly, tuning it to -2.265 (per-pixel and per-channel value). We arrived at this value by starting with -61000, which is for the unnormalized mixture-of-Gaussians summed over pixels and channels with standard deviation of 0.1 used by Emami et al. (2021)), and then increasing the target slightly to achieve a high-quality reconstruction. We also trained GENv2 on all three environments using the same Gaussian image likelihood as LOM-GENv2-G for a more fair comparison.

**GNM:** We use the official GNM PyTorch implementation provided by the authors.[5] We train GNM on the 128x128 resolution version of CLEVR6 so that we could use the default GNM hyperparameters that the authors used for the 128x128 resolution CLEVR-based environment from their paper. This worked well for CLEVR6. By default, GNM organizes its symbolic variables into a 4x4 spatial grid, which we maintained. We treat each spatial grid cell as one slot for a total of 16 slots. To train GNM on the 64x64 Objects Room and ShapeStacks scenes, we removed one downsampling and one upsampling layer from the encoder and decoder to account for the resolution being one-half that of CLEVR6. For all environments, we used the default global latent dim of 32 and $z_{\text{what}}$ dim of 64.

**NVAE:** The NVAE is trained with the open source implementation provided by the authors.[6] We changed the image likelihood from the discrete mixture of logistic distribution to the Gaussian likelihood with standard deviation of 0.7 for a fair comparison with other considered models. We also modified the default NVAE architecture for 64x64 images for use with CLEVR6 as follows. To decrease memory consumption, we remove the normalizing flows and use 16 initial channels in the encoder and decoder towers. The NVAE has three scales of spatial latents, and we extract the top-level at the lowest resolution of 12x12. Since each spatial latent group has 20 channels, this becomes a 2880-dim vector when flattened. We follow the same protocol as (Greff et al., 2019; Dittadi et al., 2021) to compute disentanglement scores for NVAE by imposing a fixed order on objects in the scene based on their attributes.

---

[5] https://github.com/JindongJiang/GNM
[6] https://github.com/NVlabs/NVAE

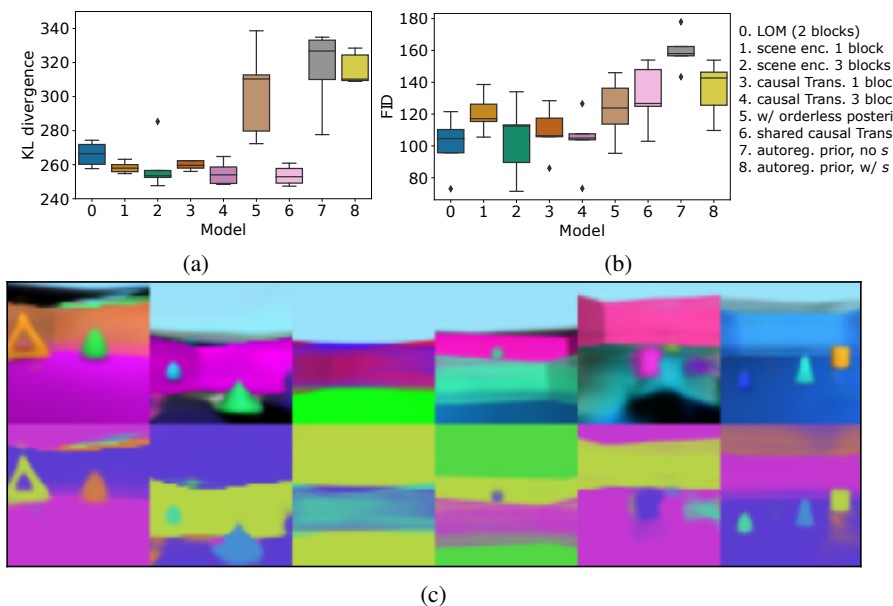

(a)                    (b)

(c)

Figure 6: **Ablation and variation results**. Box plots are shown for the five random seeds. a) Test KL for ablated models. Lower is better. b) FID scores for ablated models. Lower is better. c) Unconditional samples from the Model 8 variant, a scene-conditioned autoregressive slot prior trained to directly match an orderless posterior (using EMORL as the orderless generative model). The autoregressive prior cannot generate coherent scenes as orderless posteriors do not model any dependencies between objects. We observe similar results with the GenV2-G baseline.

**Compute:** We use a batch size of 32 for LOM-EMORL and LOM-GENv2-G. For LOM-EMORL, we split this across 8 NVIDIA A100 GPUs and train for 400K steps on Objects Room and ShapeStacks, which takes approximately 27 wall-clock hours each run. We train LOM-EMORL for 300K steps on CLEVR6 which takes 36 hours. On Objects Room and ShapeStacks, training the GENv2 baseline with 4 GPUs and a batch size of 32 takes about 20 hours to reach 500K steps (following the authors's training protocol). We train GENv2 for 300K steps on CLEVR6, which takes 24 hours. LOM-GENv2-G takes approximately a similar amount of time. GNM takes about 9 hours to reach 500K steps with a batch size of 32 on one A100 GPU for Objects Room and ShapeStacks. It takes about 12 hours to reach 500K steps on CLEVR6 on one A100 GPU. The NVAE converges quickly on CLEVR6 because we are able to use a large batch size of 256 split across 8 GPUs. We found it takes about 10-14 hours to converge in this setup depending on random seed.

## A.5 Ablation and Variation Studies

**Setup:** We train each ablation or model variant of LOM-EMORL, defined below, across five random seeds on a 48x48 resolution version of Objects Room.

**Model 0:** The baseline LOM-EMORL model, which uses 2 residual SA blocks in both the scene encoder $g_\phi$ and the causal Transformers $f_\theta$, $f'_\theta$.

**Model 1:** The number of residual SA blocks in $g_\phi$ is decreased to 1.

**Model 2:** The number of residual SA blocks in $g_\phi$ is increased to 3.

**Model 3:** The number of residual SA blocks in $f_\theta$, $f'_\theta$ is decreased to 1.

**Model 4:** The number of residual SA blocks in $f_\theta$, $f'_\theta$ is increased to 3.

**Model 5:** During LOM inference, we skip the step of computing new scale parameters for the aligned orderless slot posterior. Instead, we re-use the scale parameters estimated by the auxiliary orderless model.

**Model 6:** We share parameters between $f_\theta$ and $f'_\theta$.

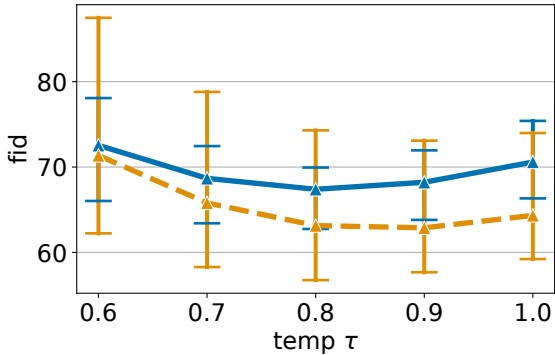

Figure 7: **FID vs. sampling temperature** ($\tau$) on Objects Room. Error bars show std. dev. across three seeds. The blue solid curve is LOM-EMORL and the orange dashed curve is LOM-GenV2-G. We use $\tau = 0.8$ for all LOM results.

**Model 7:** We consider an alternative approach to scene generation by means of an autoregressive prior. We minimize the reverse KL divergence between EMORL's orderless slot posterior and an RNN-based autoregressive slot prior, stopping gradients from passing back through the posterior. This KL is added as an auxiliary loss term to EMORL's ELBO. Note that the scene-level latent is removed in this variant and the multi-step LOM inference algorithm is not used since we are simply directly minimizing the KL between an orderless posterior and the autoregressive prior.

**Model 8:** We extend **Model 7** by adding the scene-level latent. We condition the first slot of the autoregressive slot prior on the scene-level latent $\mathbf{s}$. The scene-level reverse KL term is also added to EMORL's ELBO as an auxiliary loss.

**Results:** See Figure 6. We found that increasing the expressivity of the scene encoder $g_\phi$ by using 3 SA blocks (**Model 2**) instead of 2 (**Model 0**) slightly increased FID, whereas decreasing the number of SA blocks to 1 in $g_\phi$ (**Model 1**) and in the causal Transformers $f_\theta$, $f'_\theta$ (**Model 3**) lead to a pronounced decline in performance. Increasing the number of SA blocks to 3 for $f_\theta$, $f'_\theta$ saw negligible improvement (**Model 4**). We observed that *not* computing new order-aware scale parameters for the LOM ordered approximate slot-level posterior (**Model 5**) resulted in unfavorably high test KL and FID scores. This verifies that computing the order-aware scale parameters for the LOM approximate ordered posterior is important for learning the LOM prior. Next, we found that sharing parameters between $f_\theta$ and $f'_\theta$ severely degraded the FID (**Model 6**).

The autoregressive prior variants (**Model 7** and **Model 8**) achieve the worst FID scores and KL across all variants. However, adding the scene latent to the autoregressive prior (**Model 8**) improves the FID score relative to **Model 7**. We visualize samples from **Model 8** in Figure 6c. The orderless EMORL slot posterior (i.e., slots are independent of each other) does not provide sufficient slot dependency information to train the autoregressive prior to sequentially generate objects in a coherent way. The generated scenes appear to be mostly incoherent and have multiple walls, missing floors, and floating objects. We observed similar phenomena with GENv2, which attempts to fit an autoregressive slot prior with an orderless approximate slot posterior.

**Sampling temperature:** We plot how the Objects Room FID varies with respect to the temperature $\tau \in \{0.6, 0.7, 0.8, 0.9, 1.0\}$ in Figure 7. Decreasing the temperature to 0.8 helps samples stay within regions of high probability when sampling from the hierarchical prior. We observe a small corresponding increase in FID.

## A.6 HIERARCHICAL CLUSTERING EXAMPLES

We provide illustrative examples of how the hierarchical representation inferred by LOMs hierarchically clusters the dataset via a scene retrieval application. For a given query scene, the basic idea is to use the scene-level latent variable to first retrieve nearest-neighbor scenes based on scene-level similarities. Then, the retrieval results can be *re-ranked* by selecting a single object's slot from the

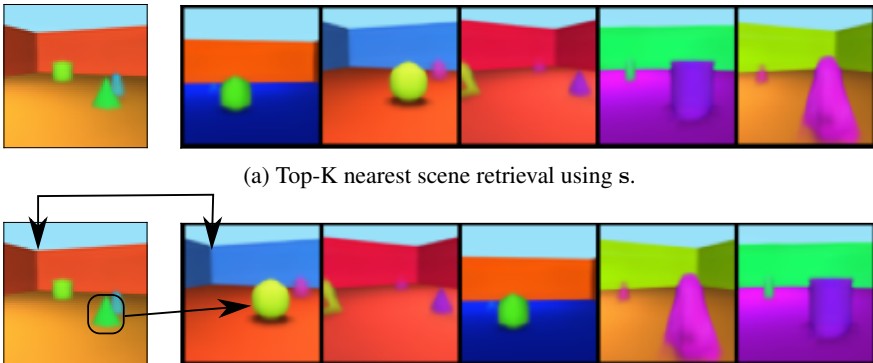

(a) Top-K nearest scene retrieval using **s**.

(b) Re-ranking results based on similarity to the green cone.

Figure 8: **Hierarchical clustering example**. The top 5 retrieved scenes based on Euclidean distance between their scene-level latent and the query scene latent (the query is shown on the left, and scenes are shown left-right in decreasing similarity rank). a) The most similar scene has similar camera perspective, wall color, object color, and object distance from the camera as the query. b) We use the slot for the green cone (boxed) to re-rank the results against the slots for each retrieved scene. The top retrieved scene has the exact same camera perspective and relative object placement (shown with arrows) as the green cone.

query and matching it against all slots of the retrieved scenes. We would expect that after re-ranking, the most similar scene to the query would also contain an object that matches the selected query object in one or more object-level attributes.

We created a database of 50K scenes stored as $(\mathbf{s}, \mathbf{z})$ pairs of vectors from the Objects Room training set. For the Euclidean distance between latent vectors to return a meaningful value, we found it necessary to first use PCA to project the 128-dim scene latents down to 6-dim vectors. We ran PCA on the database of 50K scene latents and inspected the explained variances of the principle components to select the projection dimension of 6. We select all queries from held-out samples not included in the initial 50K used to create the database. We visualize examples in Figure 8 and Figure 9.

### A.7 ADDITIONAL QUALITATIVE RESULTS

Additional examples of spherical linear interpolation (SLERP) between two random samples from the LOM scene prior are shown in Figures 10, 11, 12. To create these images, we normalize each sample **s** and then multiply by $\sqrt{D_s}$ before computing the intermediate latents.

Figure 9: **Additional hierarchical clustering examples**. Similar to Figure 8, the query is shown on the left, the top row shows the initially retrieved scenes in decreasing order of similarity, and the bottom row shows the re-ranking based on the selected slot (shown with black boxes). For ShapeStacks, we found that the initial scene-level retrieval was highly successful at finding scenes with similar camera perspective and scene background as the query.

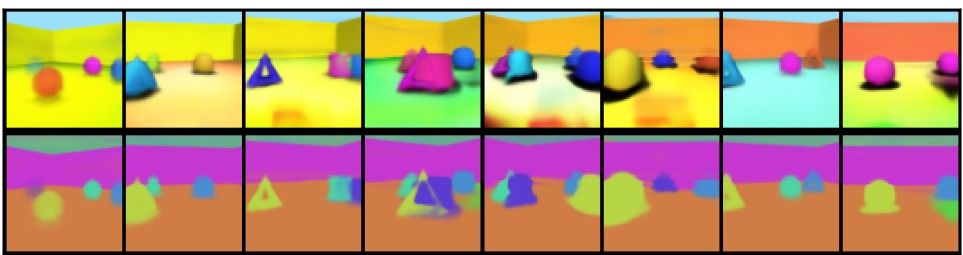

Figure 10: Objects Room SLERP between two random samples from the LOM scene prior.

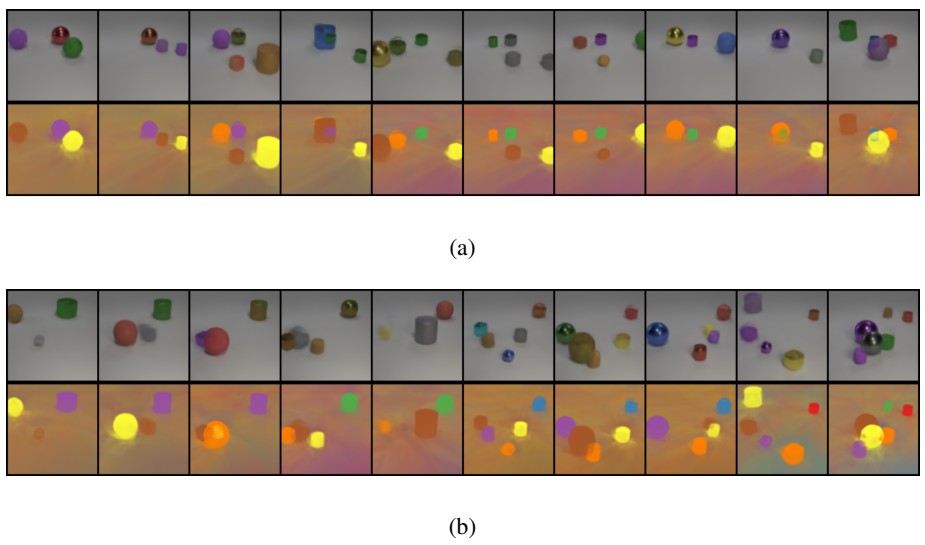

(a)

(b)

Figure 11: CLEVR6 SLERP between two random samples from the LOM scene prior.

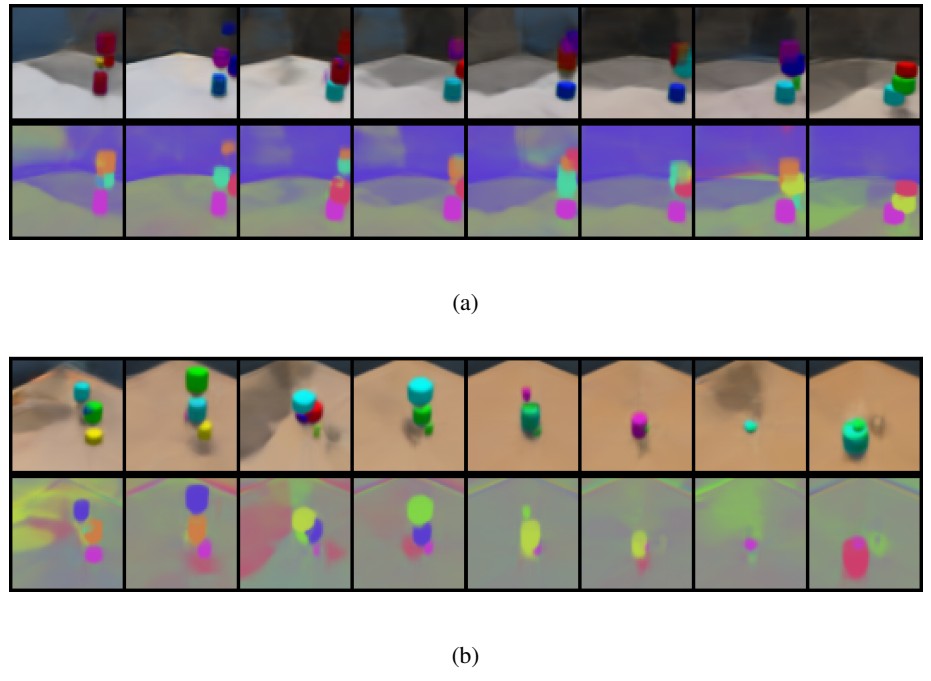

(a)

(b)

Figure 12: ShapeStacks SLERP between two random samples from the LOM scene prior.

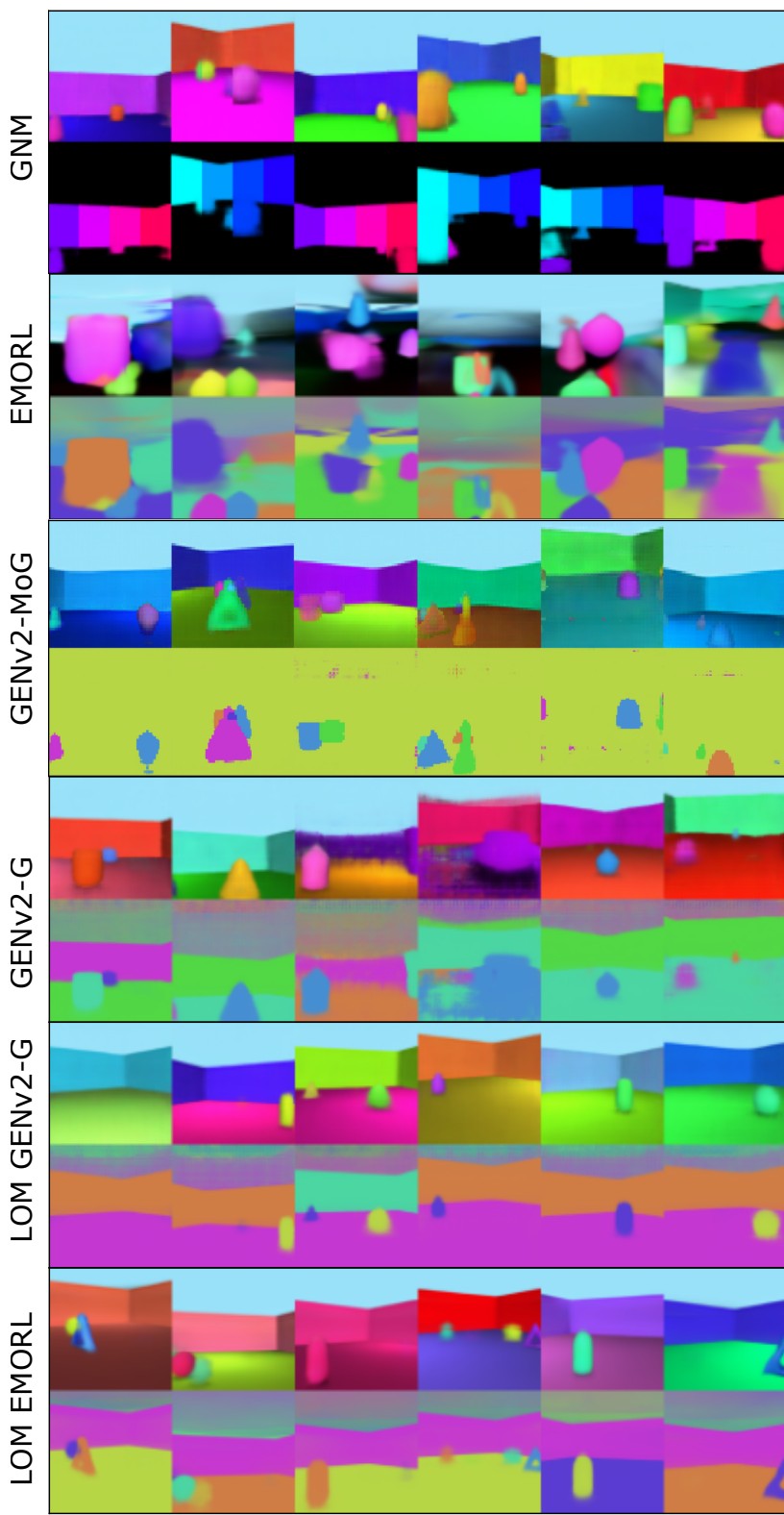

Figure 13: Objects Room unconditional samples from each generative model with both images and masks displayed.

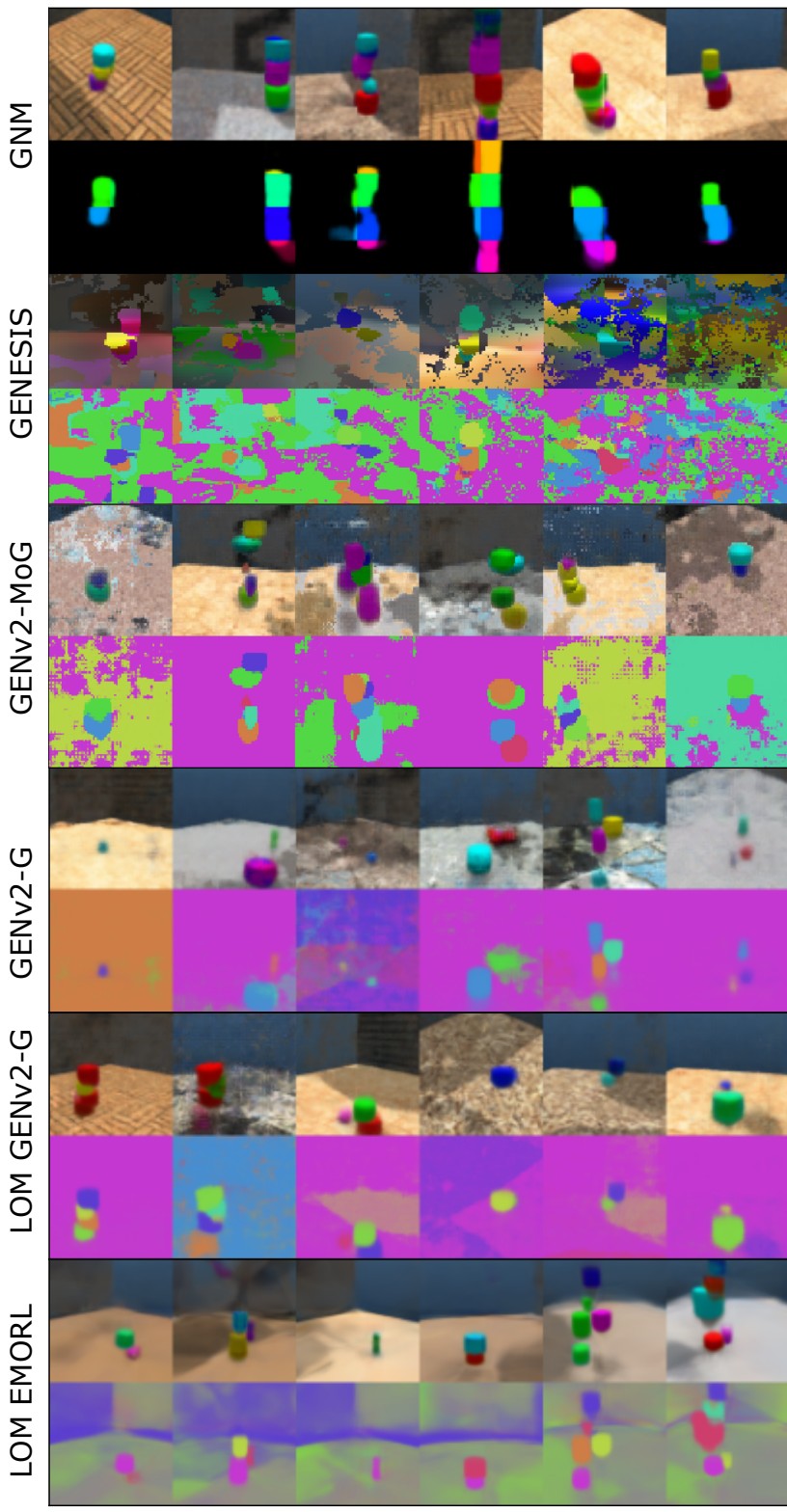

Figure 14: ShapeStacks unconditional samples from each generative model with both images and masks displayed.

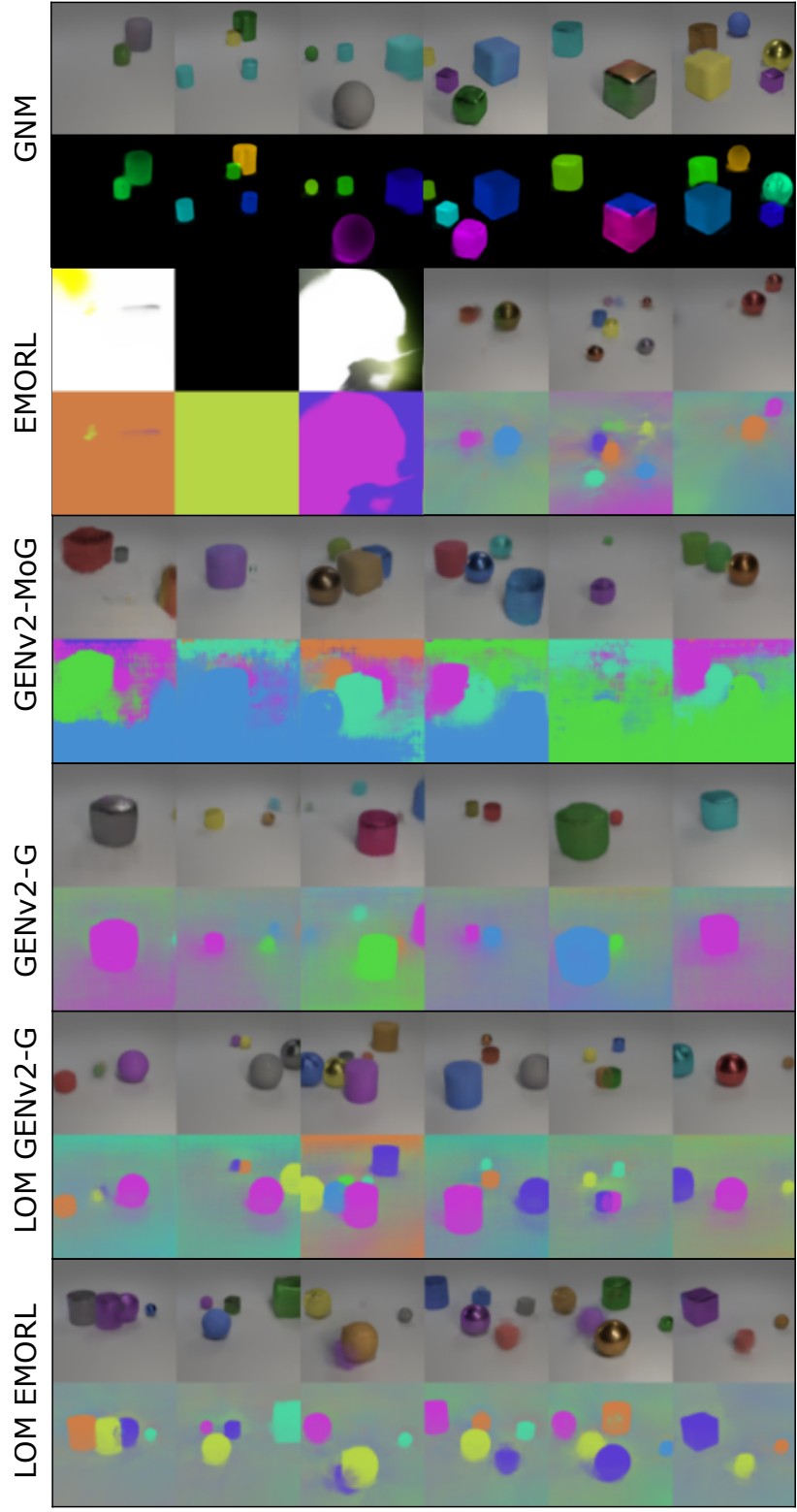

Figure 15: CLEVR6 unconditional samples from each generative model with both images and masks displayed. For EMORL, the samples shown in columns one through three are drawn from the prior of a model that scores an average ARI-FG of 96 and FID of 244. The remaining three samples are drawn from a different model trained to keep the KL low throughout training and which achieves a lower average ARI-FG of 89 but FID score of 75.

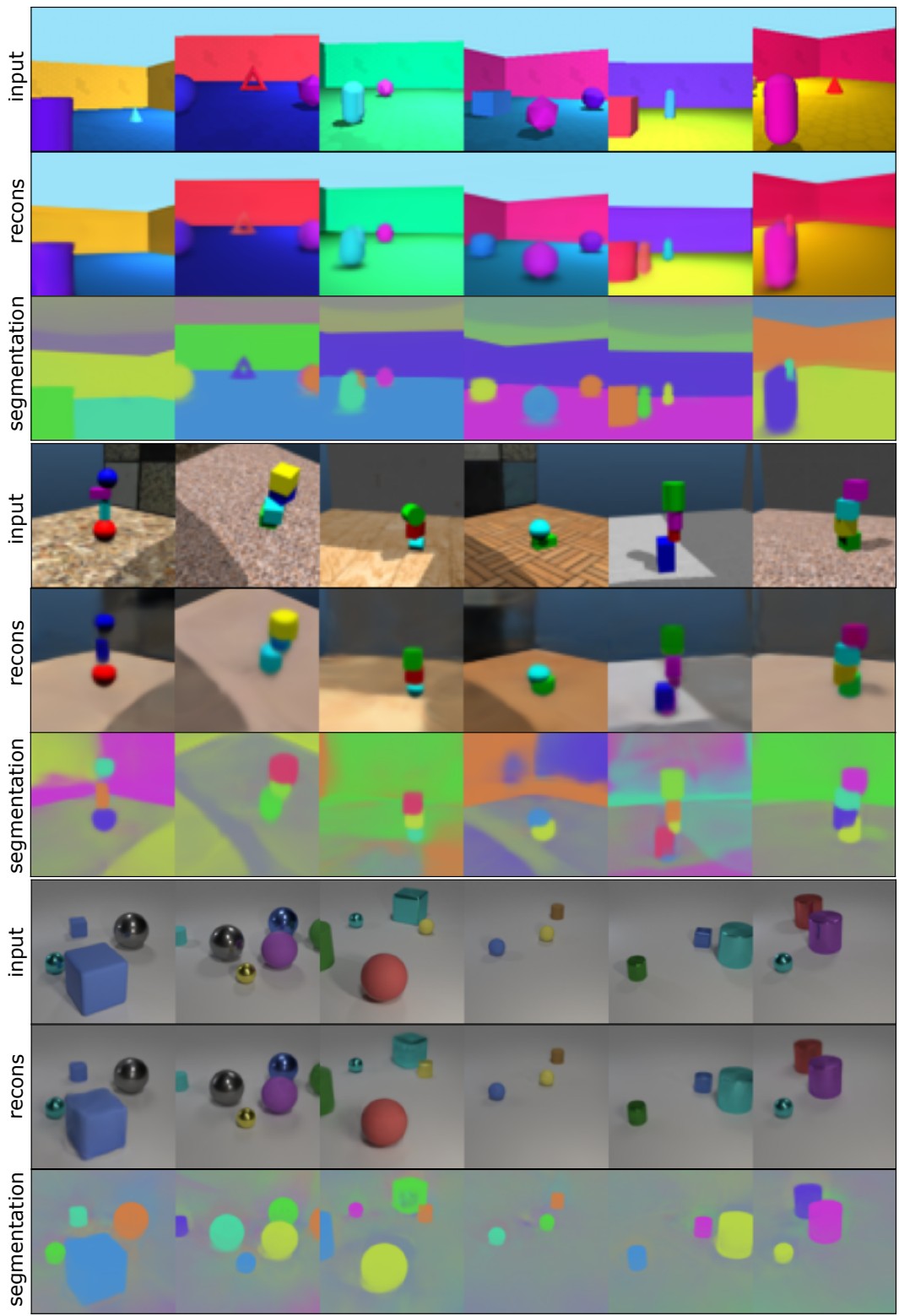

Figure 16: LOM-EMORL reconstruction and segmentation.

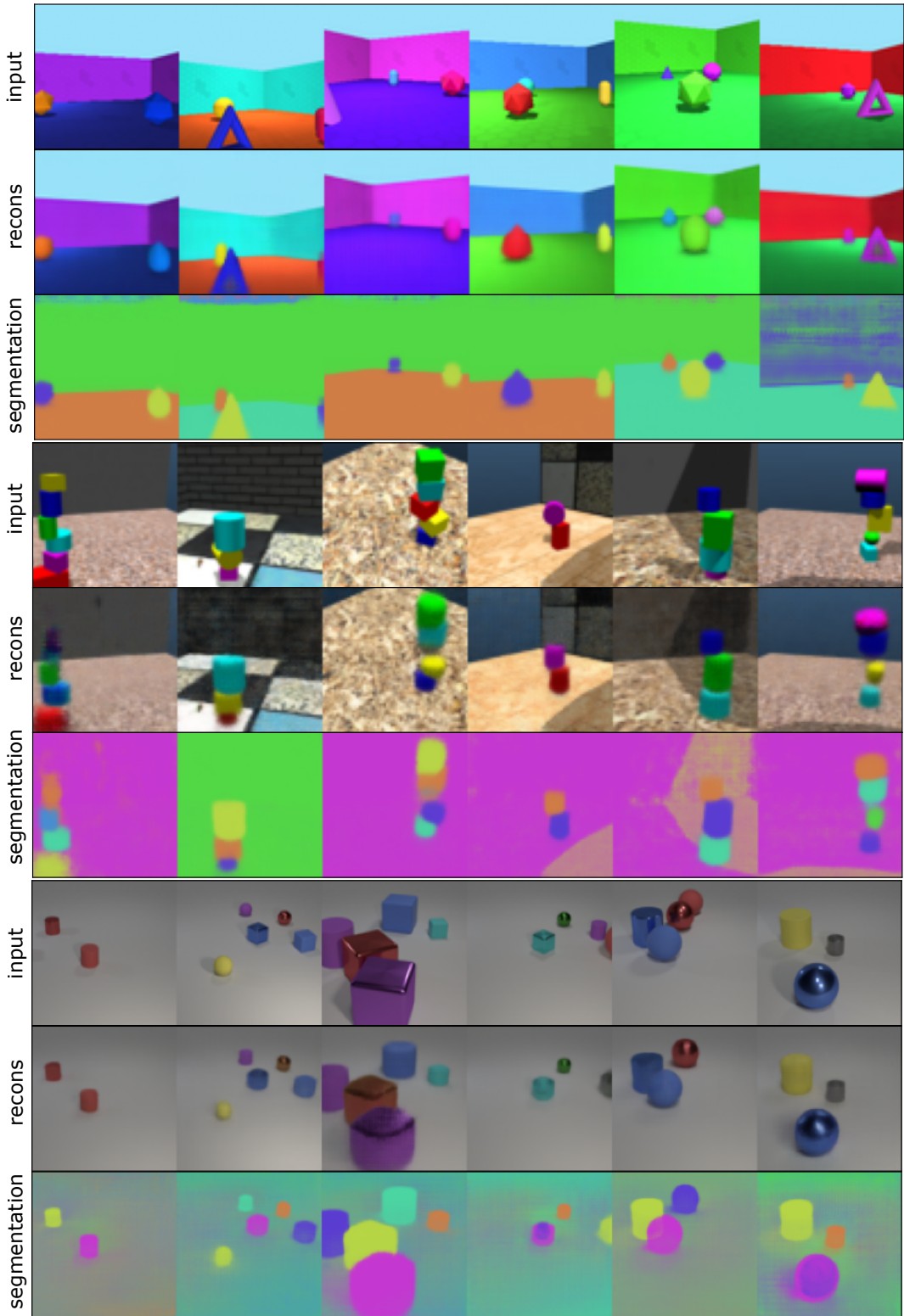

Figure 17: LOM-GenV2-G reconstruction and segmentation.

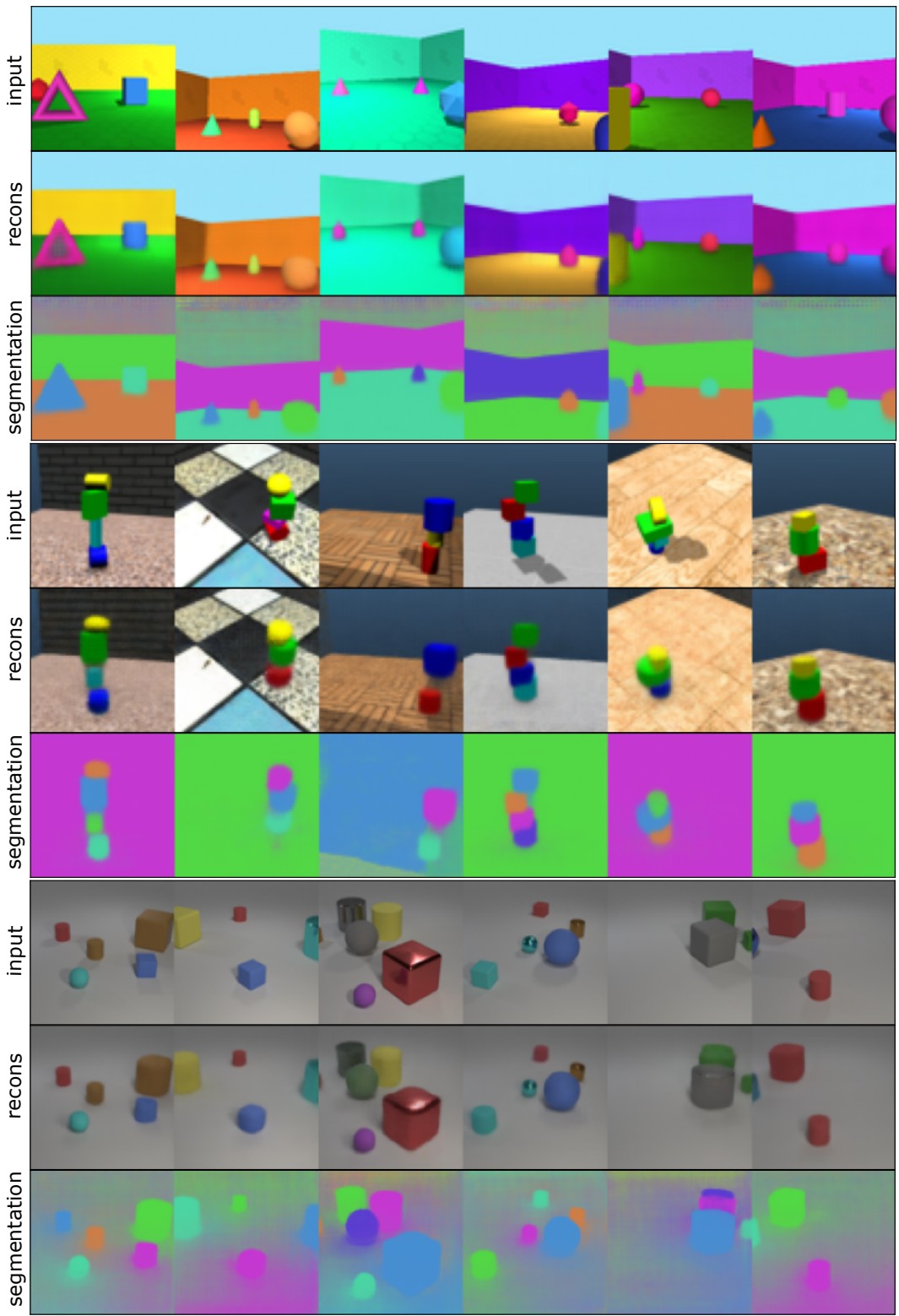

Figure 18: GENv2-G reconstruction and segmentation.

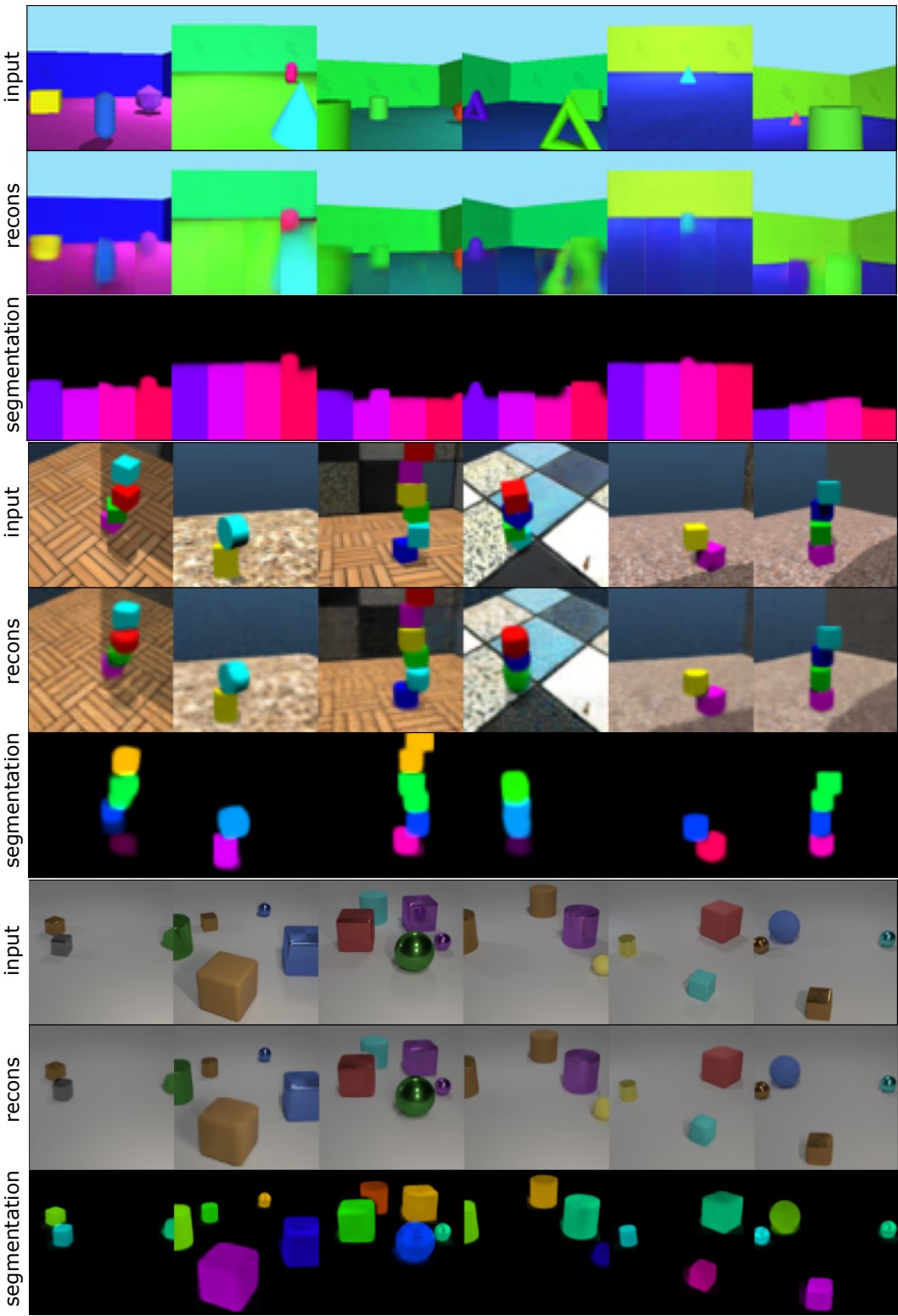

Figure 19: GNM reconstruction and segmentation. Its symbolic variables, which are bound to a spatial grid of fixed resolution, are unable to properly bind to individual objects in the Objects Room and ShapeStacks scenes. In Objects Room, the 3D shapes are segmented into strips and grouped alongside parts of the floor. The wall is erroneously not considered one of the foreground objects. In ShapeStacks, distinct blocks are often segmented together when they fall within the same grid cell.

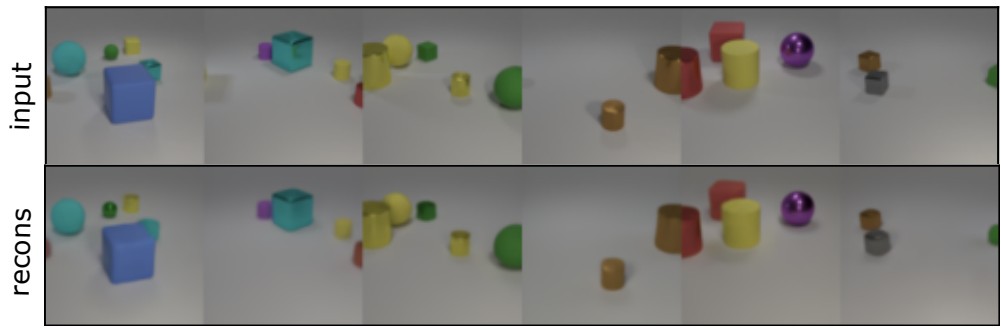

(a) Reconstruction

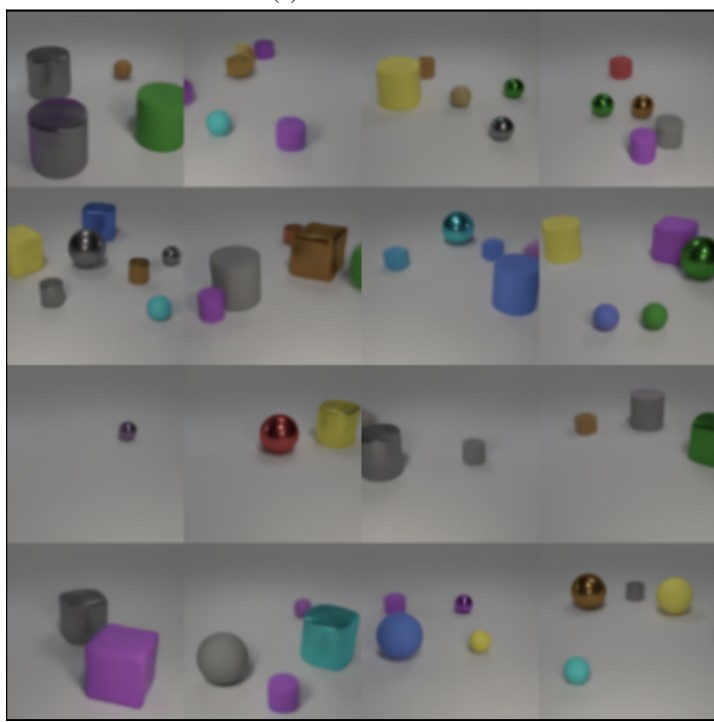

(b) Random samples

Figure 20: NVAE qualitative results.

