# OpenReview forum: "Generating Scenes with Latent Object Models"
_ICLR.cc/2022/Conference — ICLR 2022 Submitted_

### Official Review · Reviewer_4aYc · 2021-10-25

**Correctness:** 1
**Technical Novelty And Significance:** 2
**Empirical Novelty And Significance:** 2
**Recommendation:** 3
**Confidence:** 4

**Main Review:**

**Strengths**
1. The paper is well-written and the proposed method is clearly described.

2. The proposed method aims to tackle the difficulty of modeling and inferring relationships of objects, which is an important problem in object-centric representation learning of scenes.

**Weaknesses**
1. The novelty is limited as the generative model is similar to GENESIS (Engelcke et al., 2019) in that the relationships of objects are modeled by an RNN, while the inference method is similar to Slot Attention (Locatello et al., 2020).

2. The main claim/contribution that using a sequence of deterministic variables (referred as `causal sequence’ in the paper) to block dependency between scene-level random variable $s$ and slot-level random variables $z_{1:K}$ seems problematic. As shown in the generative process (Eq.1), $z_{1:K}$ still have dependences conditioned on unobserved $s$.

3. Using a causal sequence to model orders of slots is questionable. Although there are dependences between slots, the joint distribution of slot-level latent variables should be invariant under permutation because there are no causal effects among static objects.

4. Relationships between objects are not considered when estimating orderless slot posteriors. Besides the regularization of the KL divergence term, how to improve the inference of slot-level latent variables by utilizing the learned priors of object relationships is unexploited.

5. Only the scene generation performance is compared in the paper, the scene decomposition performance (e.g. ARI and AMI), which is of vital importance to evaluating object-centric learning performance, is not included. In addition, the reconstruction performance (e.g. Log-Likelihood and MSE) is also missing.

6. The proposed method is not compared with GNM (Jiang & Ahn, 2020), which also has scene latent and considers relationships among objects, and many existing relevant object-centric scene representation methods. The overall scene generation performance of the proposed method is inferior to GENESISv2 (Engelcke et al., 2021).

**Questions**
1. How to select the hyperparameter $K$?
2. How to avoid representing the same object with multiple slots?
3. Is the proposed method able to generate images containing a specific number of objects?
4. Is the proposed method able to generate images containing fewer or more objects than those used for training?


**Summary Of The Paper:**

This paper proposes a structured latent variable model, called Latent Object Models (LOM), for compositional scene generation. To generate a scene image, LOM first samples a scene-level latent variable, and then samples multiple slot-level latent variables based on a sequence of deterministic variables modeled by an autoregressive model. To infer the ordered slot-level latent variables, LOM first estimates orderless slot posteriors, and then determines the permutation of orderless slot posteriors based on a greedy matching algorithm. After learning, the model is able to generate scene images with reasonable relationships of objects. Experiments are conducted on three datasets, and the proposed method is compared with several existing methods.

**Summary Of The Review:**

This paper considers an important problem in object-centric scene representation, that is, modeling and inferring relationships among objects in a scene. However, both modeling and inference methods have heavy overlaps to existing works and, the main contribution of the paper, inserting a deterministic layer between scene-level random variable $s$ and slot-level random variables $z_{1:K}$ seems unable to block dependences among slots. The proposed method has not been compared to closely relevant methods, and the evaluation metrics for decomposition and reconstruction performance are also not fully investigated. The overall scene generation performance of the proposed method is inferior to GENESISv2.

---

> ### Author Response · Authors · 2021-11-22
> **Response to reviewer 4aYc**
>
> > *The novelty is limited as the generative model is similar to GENESIS (Engelcke et al., 2019) in that the relationships of objects are modeled by an RNN, while the inference method is similar to Slot Attention (Locatello et al., 2020)*
>
> First, we restate the novel contributions of this work:
> * LOMs are the first generative models capable of learning a hierarchy of scene-level and slot-level latent variables
> * We introduce the idea of deterministically estimating the dependencies between slots during generation using a scene-level representation
> * We introduce a novel inference algorithm for the scene and slot posterior that avoid the use of autoregressive posterior inference
>
> With respect to GENESIS:
> * LOMs possess a scene-level representation and therefore are able to hierarchically represent and generate scenes, whereas GENESIS cannot
> * The LOM prior is not autoregressive, whereas GENESIS has an autoregressive prior
> * LOMs do not need autoregressive posterior inference, whereas GENESIS relies on autoregressive posterior inference to fit dependencies between slots in its autoregressive prior
>
> With respect to Slot Attention:
> * It is unclear to us what the connection between Slot Attention and LOM inference might be that the reviewer has observed. LOM inference (see Section 3.2) is a five-step algorithm that involves imposing an ordering on an orderless slot posterior via a matching step and the ordered scene-conditioned slot prior
> * Slot Attention is not a latent variable model and hence does not infer a slot posterior
> * If the confusion is with respect to the auxiliary orderless VAE that LOMs use for inference, we highlight that we make no claims towards novelty here and simply reuse pre-existing orderless VAEs (EMORL and GENv2) in this work
>
> > *The main claim/contribution that using a sequence of deterministic variables (referred as `causal sequence’ in the paper) to block dependency between scene-level random variable $s$ and slot-level random variables $z_{1:K}$ seems problematic...*
>
> We can help clarify. We are not claiming that the deterministic variables block the dependency between the scene-level and slot-level variables.
>
> Rather, our claim is only that the deterministic variables block stochastic dependencies between the slot variables. In other words, slot $i$ is independent of slot $j$ when conditioned on the deterministic variables. Yet, both slot $i$ and slot $j$ are still dependent on the scene-level variable. Indeed, we rely heavily on the fact that the slot variables are still dependent on the scene-level variable to coherently generate scenes.
>
> > *Using a causal sequence to model orders of slots is questionable. Although there are dependences between slots, the joint distribution of slot-level latent variables should be invariant under permutation because there are no causal effects among static objects.*
>
> We acknowledge that our use of the phrase “causal sequence” can be misinterpreted to mean that we are modeling temporal causal effects, as this is perhaps the most common use. This is not the case.
>
> We are somewhat abusing the definition of “causal sequence”, which we take to mean: any element of the sequence depends on only previous elements in the sequence up to the current index. We are referring to the fact that the $i$th variable in the sequence of variables $h_1,\dots,h_K$  depends on indexes $1,\dots,i-1$ but not on any  “future” indexes $i+1,\dots,K$.
>
> > *Relationships between objects are not considered when estimating orderless slot posteriors...*
>
> As previously discussed, in this work we do not claim any novelty on the part of the auxiliary orderless VAE. It is true that they do not account for dependencies between slots.
>
> However, we show that we are able to account for these dependencies in the LOM posterior by converting the orderless posterior into an ordered posterior (Steps 3-5 of LOM inference, Section 3.2 and Figure 2).
>
> > *Missing scene decomposition performance metrics and GNM baseline*
>
> Thanks for highlighting this. We have added ARI-FG to the quantitative results. We are not familiar with AMI (it is not a standard metric from the relevant literature) and did not have room in the main text for adding MSE, but we have added Figures 16-20 to the appendix which contains qualitative visualizations of reconstruction and segmentation performance for the LOM models and key baselines. We have added the GNM baseline, which achieved much lower ARI-FG scores than the slot-based models on Objects Room and ShapeStacks.
>
> > *The overall scene generation performance of the proposed method is inferior to GENESISv2 (Engelcke et al., 2021)*
>
> Please refer to the revised paper. LOMs achieve the lowest FID score out of slot-based VAE baselines on 2 out of the 3 environments. In the third environment (Objects Room) we still demonstrate significantly improved FID scores compared to the most appropriate baseline for comparison. LOMs achieve the highest S-Acc score as well.

---

> > ### Author Response · Authors · 2021-11-23
> > **Response to Questions**
> >
> > Thanks again for your review and for these questions. We hope we can answer them to your satisfaction.
> >
> > > *How to select the hyperparameter K?*
> >
> > For our experiments, we use recommended values for K for the three environments based on prior work. In the past, this has typically been chosen to be one larger than the maximum number of objects in any scene in the dataset. Notably, one main contribution of the recently introduced GENv2 model is that it can adapt the value of K for a given scene "online".
> > We have added results for a LOM with GENv2 as the auxiliary VAE for LOM inference (LOM-GENv2-G), which means LOM-GENv2-G also shares this capability.
> >
> > > *How to avoid representing the same object with multiple slots?*
> >
> > This is handled by the scene decomposition model that LOMs use for inference. We use pre-existing models (EMORL and GENv2) to implement LOMs in this work.
> >
> > For one, we believe that the complexity of the dataset plays a large role. The three environments we consider do not contain objects with multiple parts (e.g., all three of Objects Room, ShapeStacks, and CLEVR6 have 3D shapes as objects). Therefore it is likely that the ConvNet features extracted from a scene for each object are highly self-similar. This implies that the ability of VAEs to effectively compress visual data can lead to the model learning to assign entire objects to a single slot. There are likely other inductive biases employed by these scene decomposition VAEs that contribute as well to preventing objects from being split across multiple slots.
> >
> > > *Is the proposed method able to generate images containing a specific number of objects?*
> >
> > No, the proposed method cannot control the specific number of objects in generated scenes. The distribution of the number of objects in generated scenes will be close to the underlying data distribution that the LOM is approximating.
> >
> > > *Is the proposed method able to generate images containing fewer or more objects than those used for training?*
> >
> > It is possible to vary the number of slots K at test time. We did not have time to explore increasing K but it is possible that this could result in scenes with a larger number of objects than as seen during training. This is something we will plan on investigating in the future.

---

> ### Author Response · Authors · 2021-11-29
> **Request for feedback**
>
> Dear reviewer 4aYc,
>
> Thank you again for your review of our work. As the discussion period is approaching its end, we would be grateful if you could confirm whether our responses and the additions we have made to the manuscript addressed your concerns, and let us know if any issues remain.

---

> > ### Comment · Reviewer_4aYc · 2021-11-30
> > **Feedback**
> >
> > Thank the authors for their careful revision of the paper and detailed response. The response has clarified the novelty of the proposed method. However, I still think that a more advanced inference method should be designed in order to solve the considered problem. More specifically, although slot-level random variables are independent of each other conditioned on the scene-level random variable, they are not independent when the scene-level random variable is unobserved. In other words, dependencies among slots should be considered during the inference. However, the proposed inference method heavily relies on the existing orderless VAEs, and does not fully exploit such dependencies. As the authors' response does not dispel my major concerns, I still keep the original recommendation.

---

> > > ### Author Response · Authors · 2021-11-30
> > > **Response to reviewer 4aYc**
> > >
> > > Dear reviewer 4aYc,
> > >
> > >
> > > Thanks again for your review and for your reply! We are glad that we were able to clarify concerns with the novelty of our work and hope that further clarification below on your other major concern will encourage an increase in your current score.
> > >
> > > In our response, we indicated that LOM inference **does** exploit dependencies between slots. We wrote:
> > >
> > > > *"However, we show that we are able to account for these dependencies in the LOM posterior by converting the orderless posterior into an ordered posterior (Steps 3-5 of LOM inference, Section 3.2 and Figure 2)."*
> > >
> > > Allow us to expand on this.
> > >
> > > As you emphasize, modeling dependencies between the object-level variables is essential during inference. Succinctly, the way that LOMs achieve this is by first imposing an order on the approximate orderless slot posterior (Step 4 of LOM inference) and then *computing new order-aware scale parameters for each of the $K$ slot posteriors* (Step 5 of LOM inference). Note that Step 4 of LOM inference involves permuting the order of the $K$ Gaussians in the orderless slot posterior.
> > >
> > > In Step 5 of LOM inference, we use a sequential model (an RNN or a transformer with causal attention masking is suitable) to encode the permuted slot posterior means. The order-dependent output of this sequential model, which now encodes autoregressive dependencies between the slots, is used to predict the new scale parameters. See Eq. 5 for the result---the ordered slot posterior.
> > >
> > > In detail, for $K$-way permutation $\pi$, the new scale parameter for the $\pi(k)$th Gaussian slot posterior is given by $\sigma_{\pi(k)}^{z} = w_{\phi}(h_{\pi(k)}^{'})$, with $h_{\pi(k)}^{'}  = f_\phi^{'}(o_{\pi(1):\pi(k-1)}, s)$. Here, $w_\phi$ is a linear transformation, the sequential model is $f_\phi^{'}$, $s$ is the scene-level latent variable, and $o_{\pi(1):\pi(k-1)}$ are means $1,\dots,k-1$ of the permuted orderless slot posterior. These details are in Section 3.2.
> > >
> > > Through an ablation study, we verify that computing new order-aware scale parameters for the slot posteriors successfully achieves the goal of capturing dependencies between objects. Please see our ablation study in Section **A.5**, **Model 5**. In particular, the KL ($L_{\text{slotKL}}$) is significantly higher than the baseline model, and the FID scores are much higher as well. The KL is much higher for Model 5 because, although the LOM slot prior is an *ordered* distribution and attempts to capture dependencies between objects, without computing new order-aware scale parameters the slot posterior is *orderless* and does not account for dependencies between objects.
> > >
> > > Please let us know if this has addressed your concern. We hope it is clear that the key contribution of LOM inference is *exactly* a novel algorithm for obtaining an ordered slot posterior that accounts for dependencies between objects in a scene.

---

### Official Review · Reviewer_tg1E · 2021-10-31

**Correctness:** 3
**Technical Novelty And Significance:** 3
**Empirical Novelty And Significance:** 4
**Recommendation:** 5
**Confidence:** 4

**Main Review:**

*Strenghts
This paper builds on recent methods in unsupervised object-centric segmentation and scene generation which have seen recent success in a number of synthetic image datasets. The authors give a good explanation of the problems in many of these pieces of work and motivate the need for a method that can better handle the orderless structure of the object in these scenes.  In general this paper is well written and the ideas proposed are clear and technically sound.  To my knowledge, while some of the contributions are inspired by similar ideas, the combined approach is novel and can be a useful contribution to the literature. With some exceptions noted below, the most closely related literature is compared either in the discussion and/or empirically. The experimental methodology focuses on relevant metrics, namely sampling quality, and the structure of the learned representations (e.g. structural accuracy and disentanglement), and results show for the most part that the proposed method leads to clear benefits over related work, and a large set of ablations are carried out to establish the usefulness of the different parts of the method.

*Weaknesses
1) While the key ideas are clear I find the resulting architecture, which relies on having a separately generative model, an odd choice. It makes it a bit more difficult to understand how the two models interact with each other. The new model attempts to address shortcomings of the original model, but depends on the original model to work well as it relies on its approximate posterior -- this could in some cases lead to learning limitations since gradients are not passed through the object latents of the original model. While choosing to stop the gradients is convienent (as otherwise the ordering algorithm's non-differentiability would need to be addressed), the lack of an end-to-end solution might limit the new model's benefits to propagate to learning the original model, or conversely, the limitations of the original model prevents the new model to learn e.g. robust object-centric representations.

2) The idea to deterministically generate an order to ease the task of modelling a set is not new.  For instance, Vinyals et al. show the performance benefits of finding a canonical ordering, and Zhang et al. learn to optimize these permutations. While these works don't apply it to latent set representations, I think the paper should at least discuss how some of these works relate to the proposed idea.

3) I agree with the authors that introducing a hierarchy (e.g. in the form of a global scene latent) in many of these slot-based models is a great idea. However, I would like to see more evidence of the benefits of having the prior *deterministically* produce the order of slots, as opposed to having an autoregressive slot prior. Conceptually, the burden of reasoning about the correlations between objects (in all its permutations) has now been shifted the the global prior since p(z_{1:k}|s) are now independent. This might still work for some types of scenes but I'm not sure how that would scale to scenes with larger and more complex compositions of objects. Perhaps the authors can provide some examples of what they expect the global latent to represent.

4) It's unclear what architecture is used for the likelihoood model. Is it using/sharing the same parameters as EMORL? This should be clearer in Sec. 3.

5) Wrt. the main methods compared, GENESIS and GENv2, I did not follow the explanation as to why these methods can handle richer background and textures (leading e.g. to the lower FID in Objects Room) compared to LOM. I would also like to see the resulting metrics using EMORL alone, as that would clearly show the gains of incorporating the ordered prior.

6) I don't think the error bars in Table 2 are described. It would be also interesting to see the ShapeStacks S-Acc performance across the different runs.

Some other minor comments:

- The LOM-u and LOM-s model performance was hard to find. The labelling in the text and the table in the appendix should be consistent.

- In the appendix, it is stated the Transformers have constant time dependency on the sequence length unlike RNN. While this is true if one can fully parallelize the Transformer's SA, they in general have quadratic complexity on the sequence length vs RNN's linear complexity.

**Post-rebuttal update**
While I believe there are interesting ideas in this paper, quantitative results in the paper are still mixed, with LOM-based methods mostly clearly showing its usefulness (vs e.g. GENv2-MoG) on one dataset and one metric (S-Acc). For this reason, I think the paper would benefit from having experiments using carefully designed datasets that more directly tackles the problem of modelling orderless elements using the proposed method. I would like to see the benefits on this problem compared to other ways of modelling sets. In its current form, reliance on an pretrained models complicates understanding the extent of the benefits and limitations of the method. On the point of limitations, I don't think the authors resolved my concern regarding relying on a scene-level Gaussian to capture all the structure between objects.
I appreciate the efforts of the authors to incorporate additional results and overall improving the paper, but for the reasons above  I will keep the current score.

References.

Vinyals et al: Order Matters: Sequence to sequence for sets. ICLR 2016

Zhang et al. Learning Representations of Sets through Optimized Permutations. ICLR  2019




**Summary Of The Paper:**

This work tackles the problem of learning object-centric generative models of scenes, and in particular it addresses the challenge of handling the orderless nature of objects (represented by latent variables) when learning the model's prior. Most recent models in the literature either assume an arbitrary sequential scene generation (which can hinder the model's performance), or attempt to model the orderless set of object latent variables with an autoregressive prior which can cause discontinuities during training when the allocations of object change.
The proposed method relies on incorporating (a) a scene-level prior that deterministically generates a per-object distribution ordering and (b) an ordering algorithm to match object posterior and prior distributions. By learning to generate an object order from a scene latent variable, it effectively sidesteps the problem of modelling the large equivalent class of sets. Quantitative and qualitative results show that incorporating these two ideas allows learning better generative models that can sample scenes with plausible composition of objects.

**Summary Of The Review:**

The overall ideas proposed are novel and a potentially useful contribution to the space of image VAEs. The model proposed is shown to work well on well-known multi-object datasets, and a large set of ablation studie disentangle the contributions of the different parts of their method. I'd like to see more discussions about the conceptual limitations of their ideas, as well as references to related work in finding canonical orders for set-based tasks. The authors should also address the clarity issues in the text and figures mentioned above.

---

> ### Author Response · Authors · 2021-11-22
> **Response to reviewer tg1E**
>
> > *While the key ideas are clear I find the resulting architecture, which relies on having a separately generative model,an odd choice...*
>
> Thank you for this insightful comment.  It seems plausible that future work may reveal how to remove the need to stop gradients from flowing back to the auxiliary orderless model without any negative repercussions. We also suggest that it is not obvious that we should prefer an end-to-end solution, as future investigation could reveal that although it is possible to make LOMs “end-to-end”, no real benefit is observed. Therefore, we don’t see this as a fundamental limitation but rather as an open question.
>
> The “shortcomings of the original models” is the inability of their priors to be useful for scene generation. LOMs discard these priors and depend only on the original models’ approximate orderless posterior. Any future improvements to the quality of orderless slot posterior inference can be enjoyed by LOMs.
>
> > *The idea to deterministically generate an order to ease the task of modelling a set is not new. For instance, Vinyal set al. show the performance benefits of finding a canonical ordering, and Zhang et al. learn to optimize these permutations...*
>
> This is a nice connection, thank you. We believe this is especially relevant to the proposed inference algorithm, which uses a canonical object ordering learned by the deterministic scene-conditioned function $f_\theta$ to impose an ordering on the auxiliary orderless slot posterior.  We have added the following sentence: “The algorithm shares similarities with recent work that proposes to learn a good canonical ordering for orderless sets to ease certain modeling tasks (Vinyals et al., 2016; Zhang et al., 2019).” to Section 3.2 (Inference).
>
> > *I agree with the authors that introducing a hierarchy (e.g. in the form of a global scene latent) in many of these slot-based models is a great idea. However, I would like to see more evidence of the benefits of having the prior deterministically produce the order of slots, as opposed to having an autoregressive slot prior...*
>
> We believe that LOMs will scale well to larger scenes with more complex compositions of objects for the following reasons:
> * Recall that the scene-level posterior is estimated in a hierarchical fashion. LOMs first decompose scenes into an orderless set of slots and then use a sufficiently expressive permutation-invariant function to infer the scene-level posterior. Essentially, we are introducing an object-level bottleneck to ease the compression of pixels into a single distributed representation. This key attribute of LOMs was inspired by language topic models, where the topics similarly act as a low-dimensional representation bottleneck for the document-level representation. We can efficiently compress a lot more information about scenes this way, as it is being done at a semantic level instead of at the level of individual pixels. Hence, we expect this to scale well to more complex scenes (as long as we have a way to obtain a good orderless decomposition of the scene into objects).
> * The simple sum pooling that we use in this work for global relational reasoning within the scene-level posterior can easily be replaced with more advanced methods such as graph neural networks.
> * For the scene-level prior we use a simple Gaussian distribution. More flexible parameterizations such as a mixture-of-Gaussians can be explored to fit more complex distributions for generating scenes with diverse compositions of objects.
>
> > *Wrt. the main methods compared, GENESIS and GENv2, I did not follow the explanation as to why these methods can handle richer background and textures (leading e.g. to the lower FID in Objects Room) compared to LOM. I would also like to see the resulting metrics using EMORL alone, as that would clearly show the gains of incorporating the ordered prior.*
>
> * We have added EMORL results
> * To make explicit that LOMs can be implemented (and enjoy the advantages of) any orderless slot VAE, we also added results for a LOM that uses GENv2’s orderless posterior
> * EMORL (and by extension, LOM-EMORL) appears to be more sensitive than GENv2 to the hyperparameters used to balance reconstruction and KL losses. We think one reason is that GENv2 benefits from the use of a relatively strong inductive bias that objects are spatially coherent groups of CNN features.
>   * It is also likely that investigating better schedules for trading off reconstruction and KL for EMORL will help close this gap wrt FID on environments like ShapeStacks.
> * The revised paper includes LOM-GENv2 results which achieve lower FID scores than LOM-EMORL on Objects Room and ShapeStacks (see Table 2).
> * We added a discussion on GEN and GENv2-MoG's FID scores for Objects Room in Section 4.2
>
> We have addressed issues with naming (LOM-u and LOM-s) and moved all ablations to one place in the appendix. We also clarified your point about Transformers quadratic time dependency.

---

> > ### Author Response · Authors · 2021-11-29
> > **Request for feedback**
> >
> > Dear reviewer tg1E,
> >
> > Thank you again for your review of our work. As the discussion period is approaching its end, we would be grateful if you could confirm whether our responses and the additions we have made to the manuscript addressed your concerns, and let us know if any issues remain.

---

### Official Review · Reviewer_XQMo · 2021-11-02

**Correctness:** 3
**Technical Novelty And Significance:** 3
**Empirical Novelty And Significance:** 3
**Recommendation:** 6
**Confidence:** 4

**Main Review:**

Strengths:
 - The paper addresses an important problem, namely how object-centric models can formulate a prior distribution over set-structured latents, while avoiding issues with autoregressive solutions.
 - The proposed model is novel to my knowledge. It follows the existing framework of probabilistic, object-centric modelling, but differs from existing systems in important ways.
 - The paper is generally well written.
 - The construction of the model is sensible given the design goal of eliminating the autoregressive prior from GENESIS-type models.

Weaknesses:
 - The paper lacks a clear theoretical argument as to why the autoregressive prior used in previous systems is a problem. Training autoregressive models while implicitly integrating out the factorization order has been demonstrated to work well at large scale (see e.g. the work deriving from orderless NADE, such as XLNet). It is not a priori clear to me that the solution explored here should necessarily be more effective. It might be better to introduce this issue as an open question, rather than to claim that one solution is clearly better than the other.
 - The quantitative results are mixed. LOM does not outperform previous models in terms of FID score on any of the three datasets. The main positive result is the improved structural accuracy on ShapeStacks, which is a somewhat soft metric due to its manual evaluation.
 - I am unconvinced by the qualitative results as currently presented. The interpolations in Fig. 4 do not strike me as particularly smooth, and contain inconsistent configurations with overlapping objects. Even the top ranked examples in Fig 5 b,c do not share very many commonalities with the given scene. In any case, qualitatively inspecting a single sample scene with no baseline provides too weak of a signal to support the arguments made in the text.

**Summary Of The Paper:**

The paper presents LOM, a new structured latent variable model for object-centric probabilistic modelling of scenes. It differs from previous models of this type, such as GENESIS, by avoiding the use of an autoregressive prior over slots/objects. In prior work, the use of such priors has meant that either inference needed to introduce an order among slots (e.g., by also occuring autoregressively), or the order of slots had to be implicitly integrated out during training. It is argued that either option is suboptimal. The proposed model instead models slot assignments in the prior deterministically given a scene-level variable. Inference commences largely in an orderless manner; the inferred slots are matched with their counterparts in the prior only after the fact for training.
The model is evaluated experimentally on a suite of synthetic multi-object image datasets, and compared to prior work such as GENESIS.


**Summary Of The Review:**

The paper proposes an interesting solution to the issue of formulating priors over set-structured latent variables. However, in its current state, it does not offer clear enough evidence that this solution is actually beneficial compared to previous approaches. In particular, its (potentially) beneficial properties should be more clearly demonstrated using quantitative metrics.

Post rebuttal: I have increased my score to a 6, as the authors have added additional results supporting the argument that LOMs improve the unconditional sample quality of the underlying VAE.

---

> ### Author Response · Authors · 2021-11-22
> **Response for reviewer XQMo**
>
> > *The paper lacks a clear theoretical argument as to why the autoregressive prior used in previous systems is a problem. Training autoregressive models while implicitly integrating out the factorization order has been demonstrated to work well at a large scale (see e.g. the work deriving from orderless NADE, such as XLNet). It is not a priori clear to me that the solution explored here should necessarily be more effective. It might be better to introduce this issue as an open question, rather than to claim that one solution is clearly better than the other.*
>
> Thanks for highlighting this interesting comparison with related autoregressive models. We can clarify. In our work, we address two limitations of previous slot-based VAEs for scene generation. First, they do not possess a scene-level representation. Second, we are attempting to eliminate the need for autoregressive slot posterior inference. Crucially, autoregressive posterior inference is needed to fit the dependencies between slots in an autoregressive prior. Our basic idea is to replace the autoregressive prior with an alternative scene generation prior that is not autoregressive, for which we can propose an inference algorithm that avoids directly inferring an autoregressive posterior as desired.
>
> We believe that the sentence in the second paragraph of the introduction, “Fixing the slot dependency graph vastly simplifies inference by removing the need to integrate out all ways to order the slots” may have caused confusion. Bringing up this detail early in the introduction is likely confusing so we have deleted this sentence. It is true that prior work on autoregressive models in other settings have managed to get them working well, when using both a fixed or non-fixed ordering on the random variables. The key difference between this work and models such as XLNet is that models like XLNet do not possess latent variables for which a posterior needs to be inferred. Integrating out all ways to order a sequence of N latent variables in XLNet for posterior inference would be highly expensive. Avoiding this is favorable, which is why previous slot-based VAEs like GENESIS and GENESIS-v2 use a fixed order for their autoregressive slot prior. We adopt a similar approach in our work by fixing the dependency structure of the deterministic variables, so that $h_k$ depends on variables $h_1,\dots,h_{k-1}$.
>
> > *The quantitative results are mixed. LOM does not outperform previous models in terms of FID score on any of the three datasets. The main positive result is the improved structural accuracy on ShapeStacks, which is a somewhat soft metric due to its manual evaluation.*
>
> To provide a more complete picture of the contributions of our work, we have added the FID scores for: EMORL, a GNM baseline, and a LOM implemented with GENESIS-v2 (GENv2)’s orderless posterior to our quantitative results. At the suggestion of reviewer 4aYc we have also added ARI-FG scores.
>
> After these revisions and updates, LOMs achieve the lowest FID score out of slot-based VAE baselines on ⅔ environments. In the third environment (Objects Room) where LOMs do not achieve the lowest FID score, we still demonstrate significantly improved FID scores compared to the most appropriate baseline for comparison. Please refer to the revised Section 4.2 for a more detailed breakdown of performances.
>
> > *I am unconvinced by the qualitative results as currently presented. The interpolations in Fig. 4 do not strike me as particularly smooth, and contain inconsistent configurations with overlapping objects. Even the top ranked examples in Fig 5 b,c do not share very many commonalities with the given scene. In any case, qualitatively inspecting a single sample scene with no baseline provides too weak of a signal to support the arguments made in the text.*
>
> We have updated Figure 3 in the main text to show detailed side-by-side comparisons of LOMs against key slot-based VAE baselines. We have also added a significant amount of qualitative comparisons against all baselines in the appendix, in Figures 13-20.
>
> It is true that LOM generates scenes with overlapping objects on occasion. This is not a fundamental limitation of our method. We already discussed this in our Conclusion section: “We observed that LOMs occasionally generate objects that intersect each other due to a lack of 3D understanding in the latent representations. Although unsupervised 3D scene inference is particularly difficult due to ambiguities introduced by perspective geometry, obtaining 3D object representations should improve generation quality”. Since LOM inference can be implemented with any auxiliary orderless slot posterior, any future 3D orderless slot-based models will be able to be implemented within the LOM framework.

---

> > ### Author Response · Authors · 2021-11-29
> > **Request for feedback**
> >
> > Dear reviewer XQMo,
> >
> > Thank you again for your review of our work. As the discussion period is approaching its end, we would be grateful if you could confirm whether our responses and the additions we have made to the manuscript addressed your concerns, and let us know if any issues remain.

---

> > ### Comment · Reviewer_XQMo · 2021-11-29
> > **Response**
> >
> > Thank you for your detailed response and revision. I think the additional results do a good job supporting the argument that LOMs generally improve the quality of unsupervised samples from object-centric generative models as measured by FID score. Since this was may main criticism, I am increasing my score to a 6. That said, I think reviewers 4aYc and T7nj are also right to questions whether this is the most important metric, as the practical usefulness of such models lies in their ability to extract good object-centric representations for downstream tasks, and this does not seem to improve here, at least insofar as it is measured by ARI-FG. I also remain critical of Figure 4.

---

> > > ### Author Response · Authors · 2021-11-29
> > > **To reviewer XQMo**
> > >
> > > Thanks for your review and for taking the time to examine our response and revision. And, thanks for increasing your score!
> > >
> > > You are correct that we do not demonstrate improved ARI-FG scores in this work. We would just like to point out that *this is expected*. LOMs make use of an existing model (such as EMORL and GENv2) for obtaining the orderless approximate slot posterior in the first step of LOM inference. Therefore, in our experiments, we expect the scene decomposition performance (as measured by ARI-FG) of LOM-EMORL and LOM-GENv2 to be essentially the same as EMORL and GENv2, respectively.

---

### Official Review · Reviewer_T7nj · 2021-11-06

**Correctness:** 2
**Technical Novelty And Significance:** 1
**Empirical Novelty And Significance:** 2
**Recommendation:** 3
**Confidence:** 5

**Main Review:**

1- Paper strengths:
- The proposed method does not depend on the ordering of objects in their slots
- The proposed object slots distinctly correspond to masks for each object in a scene
- Thanks to deterministic ordering of the slots, specific object slots correspond to specific semantic concepts (e.g walls, different blocks etc)


2- Paper weaknesses
- The paper needs some polishing in general both for writing and structure. Parts of the paper are written in a casual way and need to be rewritten with a more scientific perspective. Also, some of the sentences are a bit vague and hard to understand
- The Introduction section is not written well and has vague sentences; parts of it are written more similar to Related Works
- I agree that inference over all orderings of objects in a slot is hard. However, this is not well-motivated in the paper. In other words, it is not clear to me why the authors have tried to tackle the problem of integrating object orderings in a slot. This is because in practice some prior works seem to propose/fill object slots in an autoregressive fashion and produce good results at the end of the day. I think this could probably be because the models proposed in prior works do not really rely on the order of objects. As one of the experiments, the authors need to show that the results obtained from priors models are sensitive to such orderings
- Qualitatively, the produced images are not sharp enough compared to some some prior works (MONet, IODINE, some  of the missing references)
- The authors do not show any experiments for demonstrating minimal generalization of their proposed method for unseen scenes
- The main comparison of the proposed method with prior work is done quantitatively. This is not sufficient. The authors need to show results for qualitative comparison as well instead of different variations of the proposed method.

3- Additional Comments:

Abstract:
- The abstract is too wordy and not terse to the point.
- The abstract is a little bit technical to understand for general readers. It’d be better if the authors do not assume that most future readers are in their field necessarily. Examples of such technical terms: “bag-of-words assumption”, “slots are generated with an autoregressive prior”
- “known limitations”: vague, relies on authors’ prior knowledge


Introduction:
- The introduction starts by quickly talking about this specific work. It lacks a good, high-level motivation on why this problem is important.
- Pragraph2 (P2): “These models have demonstrated strong generalization and object-level disentanglement” → strong and subjective assessment. I would argue no computational model in the machine learning literature has shown strong generalization capabilities as measured against human generalization ability.
- P2 → Veerapaneni et al. does not seem to be using any of the models cited two sentences before
- “However, we understand little about how to jointly decompose scenes and sample novel
scenes from noise with slot-based models” → vague

Experiments:
- Section 4.1: I do not see an experiment in this section that clearly relate to “EXPLORING THE SCENE-LEVEL LATENT SPACE”. I was expecting some results for latent traversal here, given the section name.

References:
- Many of the references do not have the correct publication venu/information. For instance “Entity abstraction in visual model-based reinforcement learning” is an ICLR 2019 paper but is cited as an arXiv paper. Other examples: “Adam: A method for stochastic optimization”, “Auto-encoding variational bayes”, “Generative neurosymbolic machines.”, “Multi-object representation learning with iterative variational inference” etc

4- Missing References:
- White, Tom (2016): Sampling Generative Networks. Open Access Victoria University of Wellington | Te Herenga Waka. Journal contribution. https://doi.org/10.26686/wgtn.12585362.v1
- Eslami, SM Ali, Danilo Jimenez Rezende, Frederic Besse, Fabio Viola, Ari S. Morcos, Marta Garnelo, Avraham Ruderman et al. "Neural scene representation and rendering." Science 360, no. 6394 (2018): 1204-1210..
- Sitzmann, Vincent, Michael Zollhoefer, and Gordon Wetzstein. "Scene Representation Networks: Continuous 3D-Structure-Aware Neural Scene Representations." Advances in Neural Information Processing Systems 32 (2019): 1121-1132.
- Nanbo, Li, Cian Eastwood, and Robert Fisher. "Learning object-centric representations of multi-object scenes from multiple views." Advances in Neural Information Processing Systems 33 (2020).
- Crawford, Eric, and Joelle Pineau. "Learning 3D Object-Oriented World Models from Unlabeled Videos." Workshop on Object-Oriented Learning at ICML. 2020.
- Deng, Fei, et al. "Generative scene graph networks." International Conference on Learning Representations. 2020.
- Deng, Fei, Zhuo Zhi, Sungjin Ahn. Hierarchical Decomposition and Generation of Scenes with Compositional Objects,  Workshop on Object-Oriented Learning at ICML. 2020
- Chen, Chang, Fei Deng, and Sungjin Ahn. "Object-Centric Representation and Rendering of 3D Scenes." JMLR (2021).

**Summary Of The Paper:**

This work proposes a VAE-based hierarchical generative model (named Latent Object Models ) for scenes that contain multiple objects. This is done by modeling the hierarchical relationship between scenes and objects via a set of latent variables for scenes and objects. The object latent variables are a sample from their corresponding object slots. The object slots are inferred deterministically from the scene latent variable.  Having deterministic object slots makes the inference more tractable by removing the need for sequential decomposition of a scene into object slotsIn the experiments, the authors show results for random sampling of the latent variables and some qualitative results for their method and different versions of the proposed method. The authors also show quantitative comparison with some prior works

**Summary Of The Review:**

Overall, I believe the paper needs a lot more work in order to be accepted. The authors reasoning for proposing a generative model that does not depend on ordering of is not well motivated and the contributions of this work are not strong. Also, the results of the experiments the authors have performed are far from convincing, specially because they have not properly compared their method to more relevant prior works. Finally, the paper is not written well.

---

> ### Author Response · Authors · 2021-11-22
> **Response for reviewer T7nj**
>
> > *I agree that inference over all orderings of objects in a slot is hard. However, this is not well-motivated in the paper. In other words, it is not clear to me why the authors have tried to tackle the problem of integrating object orderings in a slot. This is because in practice some prior works seem to propose/fill object slots in an autoregressive fashion and produce good results at the end of the day. I think this could probably be because the models proposed in prior works do not really rely on the order of objects. As one of the experiments, the authors need to show that the results obtained from priors models are sensitive to such orderings*
>
> In our work, we address two limitations of previous slot-based VAEs for scene generation. First, they do not possess a scene-level representation. Second, we are attempting to eliminate the need for autoregressive slot posterior inference. Crucially, autoregressive posterior inference is needed to fit the dependencies between slots in an autoregressive prior. Our basic idea is to replace the autoregressive prior with an alternative scene generation prior that is not autoregressive, for which we can propose an inference algorithm that avoids directly inferring an autoregressive posterior as desired.
>
> We believe that the sentence in the second paragraph of the introduction, “Fixing the slot dependency graph vastly simplifies inference by removing the need to integrate out all ways to order the slots” may have caused confusion. Bringing up this detail early in the introduction is likely confusing so we have deleted this sentence. We simply are referring here to the fact that previous methods such as GENESIS and GENESIS-v2 use a fixed order for the autoregressive slot prior. We adopt a similar approach in our work by fixing the dependency structure of the deterministic variables, so that $h_k$ depends on variables $h_1,\dots,h_{k-1}$.
>
>
> > *Qualitatively, the produced images are not sharp enough compared to some prior works (MONet, IODINE,some of the missing references)*
>
> We have added a LOM model based on GENv2 (LOM-GENv2-G), which qualitatively achieves sharper generation on Objects Room and ShapeStacks than LOM-EMORL and improves on its FID scores (see Figure 3, 4, and in the appendix, Figures 13-15, Figure 17).
> MONet and IODINE are not capable of structured scene generation so we consider this comparison is not appropriate. Alternatively, we can consider GENESIS as it is a “generative” version of MONet. Its scene generation performance on ShapeStacks is significantly worse than LOM-EMORL and LOM-GENv2. Note that IODINE has an i.i.d. prior similar to EMORL. In the revised paper, we include qualitative and quantitative results for EMORL and verify that LOM-EMORL achieves significantly better FID scores than EMORL.
>
> > *The authors do not show any experiments for demonstrating minimal generalization of their proposed method for unseen scenes*
>
> The extent of generalization that we can expect from models purporting to do scene density estimation, such as LOMs, is generating novel scenes not present in the training data by interpolating between two scenes. This can be demonstrated by traversing the latent space between two samples from the scene prior.
>
> We did provide an example traversal of the scene-level latent space in Figure 4 (note that we updated this figure with a smaller qualitative example from only Objects Room to save on space in the revised paper). More latent traversals are also shown in Figure 10-12.
>
> > *The main comparison of the proposed method with prior work is done quantitatively. This is not sufficient.*
>
> We have improved the qualitative evaluation to include detailed side-by-side visual comparisons against key slot-based VAE baselines. Figure 3 has been updated and we have updated the text in Section 4.1 to discuss the added comparisons. We have also added numerous additional qualitative comparisons in the appendix in Figures 13-15.
>
> > *Section 4.1: I do not see an experiment in this section that clearly relate to “EXPLORING THE SCENE-LEVEL LATENT SPACE”.*
>
> We have these results. The results are in Figure 4 which shows the latent traversal of the scene-level latent variable. We have changed the name of this section to be “Qualitative Evaluation” so that it more generally refers to all qualitative evaluations. We also show additional traversals in Figures 10-12.
>
>
> > *Bibliography*
>
> Thanks for pointing out the missing venues and information in our bibliography. We will address this before the camera-ready deadline.
> We have added references to relevant mentioned papers in appropriate locations in the revised version.

---

> > ### Comment · Reviewer_T7nj · 2021-11-29
> > **Thank you**
> >
> > Dear authors, thank you so much for the time you took to revise the paper with more details and more qualitative results in addition to your attempts to address the reviewers' concerns and/or clarify their confusions. Your response clarified some of the things for me and I changed my recommendation for the paper accordingly.
> >
> > However, my main concern is that you may seem to be trying to solve a problem that does not seem to be very important to the community that I mention below:
> >
> > 1- There are other works (e.g. Scene Representation Networks, Neural Scene Representation and Rendering, BlockGAN) that do not rely on an autoregressive slot priors and produce good results. How would you compare your work to those works? If your work is not relevant to those other works, what is the advantage of your work compared to them?
> >
> > 2- I'm not entirely sure what you mean by "they do not possess a scene-level representation". As far as I know, there are lots of prior works (I pointed out some in the missing citations) that learn scene-level representations and generate scenes conditioned on such representation. I'd appreciate if you can try to provide some clarification on both of these concerns.
> >
> > Additionally, it would have been great if you had included some results for an experiment to evaluate the generalization limits of your model. For instance, I'm curious to know whether your method be able to do reconstruction for a scene with more objects that your model seen during training.

---

> > > ### Author Response · Authors · 2021-11-29
> > > **Response to questions**
> > >
> > > Dear Reviewer T7nj,
> > >
> > > We would be happy to clarify the differences and advantages of Latent Object Models with respect to these other models.
> > >
> > > ### Question 1
> > >
> > > Neural Scene Representation and Rendering introduces GQN, a VAE that is designed for *conditional* scene generation. Conditional scene generation involves learning to approximate the conditional likelihood $p(D | c)$ where $D$ is the dataset and $c$ is some auxiliary information (e.g., a context image and/or camera pose). By contrast, LOMs are designed for *unconditional* scene generation---which is enabled by learning $p(D)$, which is more difficult.  Additionally, GQNs represent scenes with a single latent code. The ROOTS model of Chen et al. 2021 has been proposed as a 3D *object-centric* extension of GQN. However, it still only learns a conditional likelihood and uses the same object-centric scene representation as the GNM baseline discussed in our paper and therefore suffers from the same limitations as GNM.
> > >
> > > Scene Representation Networks (SRNs) are auto-decoder models for synthesizing scene views from a query camera pose. They resemble a deterministic version of GQNs---both GQNs and SRNs require a multi-view dataset for training, while LOMs do not. SRNs are not a proper generative model; they cannot handle uncertainty and do not try to learn $p(D)$--see their discussion on the d(eterministic)GQN baseline in their appendix (Section 5). They are also limited to handling datasets of single object images without backgrounds. To handle multiple instances of a single object class, they must employ an additional hypernetwork. Unlike VAEs like LOMs, their auto-decoder framework obtains latent codes for a scene via an optimization process rather than via a single forward pass.  Finally, SRNs are not able to learn an object-centric scene representation, one of the key desiderata of our work.
> > >
> > > We agree that these discussions could be helpful for obtaining a better understanding of the positioning of this work and we will add an expanded related work section to the appendix for the camera-ready version.
> > >
> > > We have discussed BlockGAN and related GAN models in the last paragraph of our related work. Such *likelihood-free* deep generative models are unable to support *scene decomposition*, unlike LOMs. This is because they *implicitly* learn $p(D)$, whereas our goal is to learn an *explicit* approximation to $p(D)$.
> > >
> > > ### Question 2
> > >
> > > The line “do not possess a scene-level representation” in the second paragraph of the introduction is written in reference to previous likelihood-based multi-object VAEs. These are the most relevant previous models suitable for learning to approximate the data-generating process for a dataset $D$ of multi-object scenes (e.g., learning $p(D)$). However, they are limited in their ability to achieve this goal because they only represent the scene at the level of individual objects. We highlight this in Table 1. This line does not refer to other models such as BlockGAN, etc.
> > >
> > > ### On generalization
> > >
> > > We agree that generalization is an important advantage of this class of models. First, we emphasize that we make no claims of *improving* the generalization capabilities of object-centric generative models, and therefore placed less priority on including these results. Second, unfortunately, we did not have time to run these extra experiments for the paper revision. However, we can point to prior work that has evaluated the generalization of slot-based models to more objects than as seen during training:
> > > * Emami et al., 2021 (EMORL) - Section **4.2** “Systematic generalization”
> > > * Greff et al., 2019 (IODINE) -  Section **4.2** “Generalization”
> > > * Dittadi et al., 2021 (compares MONet, Slot Attention, and GENESIS) - Section **4.4**, Figure **8**
> > >
> > > All three experiments demonstrate successful systematic generalization to larger numbers of objects.  Only a slight drop in reconstruction and segmentation performance is observed. Note that LOM inference leverages such orderless slot-based VAEs and hence enjoys their respective advantages (including the ability to generalize to more objects than as seen during training).
> > >
> > >
> > > References
> > >
> > > 1. Chen, Chang, Fei Deng, and Sungjin Ahn. "Object-Centric Representation and Rendering of 3D Scenes." JMLR (2021)
> > > 2. Emami, Patrick, Pan He, Sanjay Ranka, Anand Rangarajan. “Efficient Iterative Amortized Inference for Learning Symmetric and Disentangled Multi-Object Representations” ICML 2021.
> > > 3. Greff, Klaus, Raphaël Lopez Kaufman, Rishabh Kabra, Nick Watters, Christopher Burgess, Daniel Zoran, Loic Matthey, Matthew Botvinick, Alexander Lerchner. “Multi-Object Representation Learning with Iterative Variational Inference”. ICML 2019.
> > > 4. Dittadi, Andrea, et al. "Generalization and robustness implications in object-centric learning." arXiv preprint arXiv:2107.00637 (2021).

---

### Author Response · Authors · 2021-11-22
**Main response**

We would like to thank all the reviewers for their time and for providing us with helpful feedback and suggestions to improve this work. We have strengthened the paper based on your comments and kindly ask reviewers to take a careful look at the revised PDF.
In summary:

* Experiments
  * Added results for the GNM baseline and for EMORL -- Table **2**
  * Added results for a Latent Object Model with GENESIS-v2 as the auxiliary VAE used for LOM inference with Gaussian image likelihood (LOM-GENv2-G) --  Table **2**, Figure **3**
  * Added the ARI-FG metric -- Table **2**
  * Split the quantitative metrics across two tables for better space management, which allowed us to move the disentanglement scores into the main text -- Table **3**
  * Moved the short subsection on ablation studies from the experiment results to the appendix, where we have more space to carefully detail all ablations/variations -- Appendix Section **A.5**, Figure **6c**
  * Moved the hierarchical clustering/scene retrieval demonstration to the appendix to create needed space -- Appendix Section **A.6**
  * Improved the qualitative results by showing side-by-side comparisons against key slot-based VAE baselines -- Figure **3**
  * Replaced the old SLERP figure with a new one that takes less space -- Figure **4**
  * Updated the discussion of the qualitative results to include mention of the comparisons between models in the new Figure 3 -- Section **4.1**
  * Updated the discussion of the quantitative results to mention the added EMORL, GNM, and LOM-GENv2-G models -- Section **4.2**
  * Updated the appendix with many more qualitative examples (reconstruction and segmentation, and random samples) and side-by-side comparisons -- Appendix Section **A.7** Figures **13-19**
* References
   * Added relevant references suggested by reviewers in the appropriate places -- [Vinyals et al., 2016; Zhang et al., 2019; Deng et al., 2021; Nanbo et al., 2020; Chen et al., 2020]
* Clarity
  * Removed or revised vague statements in the abstract and introduction to attempt to help clarify precisely the motivation and our contributions -- Section **1**

Other minor improvements:
* Visualization of the neural architectures used for the LOM generative model -- Appendix Section **A.1**, Figure **5**
* Added a study on FID vs. sampling temperature $\tau$ -- Appendix Section **A.5**, Figure **7**

We will reply to the reviewer feedback and questions directly. We hope that the revised paper and detailed responses bring clarity to any misunderstandings about the problem we are tackling and the contributions of our work. If our replies are found to be satisfactory, we kindly ask that reviewers increase their scores to reflect their new opinion of the work.

---

### Decision · Program_Chairs · 2022-01-20

**Decision:**

Reject

**Comment:**

This paper proposes a VAE-based hierarchical generative model (Latent Object Model) to model scenes with multiple objects.
The paper would benefit from a substantial revision to improve text quality and clarity.
The experiments lack proper quantitative baselines and imputations; and the overall results are quite underwhelming relative to existing models.